# OFFLINE REINFORCEMENT LEARNING FROM HETERO-SKEDASTIC DATA VIA SUPPORT CONSTRAINTS

## ABSTRACT

Offline reinforcement learning (RL) learns policies entirely from static datasets, thereby avoiding the challenges associated with online data collection. Practical applications of offline RL will inevitably require learning from datasets where the variability of demonstrated behaviors changes non-uniformly across the state space. For example, at a red light, nearly all human drivers behave similarly by stopping, but when merging onto a highway, some drivers merge quickly, efficiently, and safely, while many hesitate or merge dangerously. Both theoretically and empirically, we show that typical offline RL methods, which are based on distribution constraints fail to learn from data with such non-uniform variability, due to the requirement to stay close to the behavior policy **to the same extent** across the state space. Ideally, the learned policy should be free to choose **per state** how closely to follow the behavior policy to maximize long-term return, as long as the learned policy stays within the support of the behavior policy. To instantiate this principle, we reweight the data distribution in conservative Q-learning (CQL) to obtain an approximate support constraint formulation. The reweighted distribution is a mixture of the current policy and an additional policy trained to mine poor actions that are likely under the behavior policy. Our method, CQL (ReDS), is simple, theoretically motivated, and improves performance across a wide range of offline RL problems in Atari games, navigation, and pixel-based manipulation.

## 1 INTRODUCTION

Recent advances in offline RL (Levine et al., 2020; Lange et al., 2012) hint at exciting possibilities in learning high-performing policies, entirely from offline datasets, without requiring dangerous (García & Fernández, 2015) or expensive (Kalashnikov et al., 2018) active interaction with the environment. Analogously to the importance of data diversity in supervised learning (Deng et al., 2009), the practical benefits of offline RL depend heavily on the *coverage* of behavior in the offline datasets (Kumar et al., 2022). Intuitively, the dataset must illustrate the consequences of a diverse range of behaviors, so that an offline RL method can determine what behaviors lead to high returns, ideally returns that are significantly higher than the best single behavior in the dataset.

We posit that combining many realistic sources of data can provide this kind of coverage, but doing so can lead to the variety of demonstrated behaviors varying in highly non-uniform ways across the state space, i.e. heteroskedastic datasets. For example, a dataset of humans driving cars might show very high variability in driving habits, with some drivers being timid and some more aggressive, but remain remarkably consistent in critical states (e.g., human drivers are extremely unlikely to swerve in an empty road or drive off a bridge). A good offline RL algorithm should combine the *best* parts of each behavior in the dataset – e.g., in the above example, the algorithm should produce a policy that is *as good as the best human in each situation*, which would be better than *any* human driver overall. At the same time, the learned policy should not attempt to extrapolate to novel actions in subset of the state space where the distribution of demonstrated behaviors is narrow (e.g., the algorithm should not attempt to drive off a bridge). How effectively can current offline RL methods selectively choose on a *per-state* basis how closely to stick to the behavior policy?

Most existing methods (Kumar et al., 2019; 2020; Kostrikov et al., 2021b;a; Wu et al., 2019; Fujimoto et al., 2018a; Jaques et al., 2019) constraint the learned policy to stay close to the behavior policy, so-called "distributional constraints". Our first contribution consists of empirical and theoretical evidence demonstrating that distributional constraints are insufficient when the heteroskedas-

(a) Variability in behavior depends on state  (b) AWR is stuck  (c) CQL is stuck  (d) CQL (ReDS) succeeds

Figure 1: **Failure mode of distributional constraints.** In this navigation task, an offline RL algorithm must find a path from the start state to the goal state as indicated in (a). The offline dataset provided exhibits non-uniform coverage at different state, e.g., in the state marked as "B" located in a wide room has more uniform action distribution, whereas the states in the narrow hallways exhibit a more narrow action distribution. This is akin to how the behavior of human drivers varies in certain locations ("B"), but is very similar in other situations ("A"). To perform well, an algorithm must stay close to the data in the hallways ("A"), but deviate significantly from the data in the rooms ("B"), where the data supports many different behaviors (most are not good). AWR and CQL become stuck because they stay too close to the bad behavior policy in the rooms, e.g. the left and right arrows near State B in Fig (b) and (c). Our method, CQL (RedS), learns to ignore the bad behavior policy action in state B and prioritizes the good action, indicated by the downward arrow near State B in (d).

ticity of the demonstrated behaviors varies non-uniformly across states, because the strength of the constraint is state-agnostic, and may be overly conservative at some states even when it is not conservative enough at other states. We also devise a measure of heteroskedasticity that enables us to determine if certain offline datasets would be challenging for distributional constraints.

Our second contribution is a simple and theoretically-motivated observation: distribution constraints against a *reweighted* version of the behavior policy give rise to support constraints. That is, the return-maximization optimization process can freely choose state-by-state how much the learned policy should stay close to the behavior policy, so long as the learned policy remains within the data support. We show that it is particularly convenient to instantiate this insight on top of conservative Q-learning (CQL) (Kumar et al., 2020), a recent offline RL method. The new method, CQL (ReDS), only changes minimally the form of regularization, design decisions employed by CQL and inherits existing hyper-parameter values. CQL (ReDS) attains better performance than recent distribution constraints methods on a variety of tasks with more heteroskedastic distributions.

## 2 PRELIMINARIES ON DISTRIBUTIONAL CONSTRAINTS OFFLINE RL

The goal in offline RL is find the optimal policy in a Markov decision process (MDP) specified by the tuple $\mathcal{M} = (\mathcal{S}, \mathcal{A}, T, r, \mu_0, \gamma)$. $\mathcal{S}, \mathcal{A}$ denote the state and action spaces. $T(\mathbf{s}'|\mathbf{s}, \mathbf{a})$ and $r(\mathbf{s}, \mathbf{a})$ represent the dynamics and reward function. $\mu_0(s)$ denotes the initial state distribution. $\gamma \in (0, 1)$ denotes the discount factor. The goal is to learn a policy that maximizes the return, denoted by $J(\pi) := \frac{1}{1-\gamma}\mathbb{E}_{(\mathbf{s}_t, \mathbf{a}_t) \sim \pi}[\sum_t \gamma^t r(\mathbf{s}_t, \mathbf{a}_t)]$. We must find the best possible policy while only having access to an offline dataset of transitions collected using a behavior policy $\pi_\beta$, $\mathcal{D} = \{(\mathbf{s}, \mathbf{a}, r, \mathbf{s}')\}$.

**Offline RL via distributional constraints**. Most offline RL algorithms regularize the learned policy $\pi$ from querying the target Q-function on unseen actions (Fujimoto et al., 2018a; Kumar et al., 2019), either implicitly or explicitly. For our theoretical analysis, we will abstract the behavior of distributional constraint offline RL algorithms into a generic formulation following Kumar et al. (2020). As shown in Equation 1, we consider the problem where we must maximize the return of the learned policy $\pi$ (in the empirical MDP) $\widehat{J}(\pi)$, while also penalizing the divergence from $\pi_\beta$:

$$\max_\pi \ \mathbb{E}_{\mathbf{s} \sim \widehat{d}^\pi} \left[ \widehat{J}(\pi) - \alpha D(\pi, \pi_\beta)(\mathbf{s}) \right] \quad \text{(generic distributional constraint)} \quad (1)$$

where $D$ denotes a divergence between the learned policy $\pi$ and the behavior policy $\pi_\beta$ at state $\mathbf{s}$.

**Conservative Q-learning.** (Kumar et al., 2020) enforces the distributional constraint on the policy *implicitly*. To see why this is the case, consider the CQL objective, which consists of two terms:

$$\min_\theta \ \alpha \underbrace{(\mathbb{E}_{\mathbf{s} \sim \mathcal{D}, \mathbf{a} \sim \pi}[Q_\theta(\mathbf{s}, \mathbf{a})] - \mathbb{E}_{\mathbf{s}, \mathbf{a} \sim \mathcal{D}}[Q_\theta(\mathbf{s}, \mathbf{a})])}_{\mathcal{R}(\theta)} + \frac{1}{2}\mathbb{E}_{\mathbf{s}, \mathbf{a}, \mathbf{s}' \sim \mathcal{D}}\left[\left(Q_\theta(\mathbf{s}, \mathbf{a}) - \mathcal{B}^\pi \bar{Q}(\mathbf{s}, \mathbf{a})\right)^2\right], \quad (2)$$

where $\mathcal{B}^\pi \bar{Q}(\mathbf{s}, \mathbf{a})$ is the Bellman backup operator applied to a delayed target Q-network, $\bar{Q}$: $\mathcal{B}^\pi \bar{Q}(\mathbf{s}, \mathbf{a}) := r(\mathbf{s}, \mathbf{a}) + \gamma \mathbb{E}_{\mathbf{a}' \sim \pi(\mathbf{a}'|\mathbf{s}')}[\bar{Q}(\mathbf{s}', \mathbf{a}')]$. The second term (in blue) is the standard TD error (Lillicrap et al., 2015; Fujimoto et al., 2018b; Tuomas Haarnoja & Levine, 2018). The first term $\mathcal{R}(\theta)$ (in red) attempts to prevent overestimation in the Q-values for out-of-distribution (OOD)

actions by minimizing the Q-values under a distribution $\mu(\mathbf{a}|\mathbf{s})$, which is automatically chosen to pick actions with high Q-values $Q_\theta(\mathbf{s}, \mathbf{a})$, and counterbalances by maximizing the Q-values of the actions in the dataset. Kumar et al. (2020) show that Equation 2 gives rise to a pessimistic Q-function that modifies the optimal Q function by the ratios of densities, $\pi(\mathbf{a}|\mathbf{s})/\pi_\beta(\mathbf{a}|\mathbf{s})$ at a given state-action pair $(\mathbf{s}, \mathbf{a})$. More formally, the Q-function obtained after one iteration of TD-learning is given by:

$$Q_\theta(\mathbf{s}, \mathbf{a}) := \mathcal{B}^\pi \bar{Q}(\mathbf{s}, \mathbf{a}) - \alpha \left[ \frac{\pi(\mathbf{a}|\mathbf{s})}{\pi_\beta(\mathbf{a}|\mathbf{s})} - 1 \right]. \tag{3}$$

The Q function is unchanged only if the density of the learned policy $\pi$ matches that of the behavior policy $\pi_\beta$. Otherwise, for state-action pairs where $\pi(\mathbf{a}|\mathbf{s}) < \pi_\beta(\mathbf{a}|\mathbf{s})$, Eq. 3 increases their Q values and encourages the policy $\pi$ to assign more mass to the action. Vice versa, if $\pi(\mathbf{a}|\mathbf{s}) > \pi_\beta(\mathbf{a}|\mathbf{s})$, Eq. 3 encourages the policy $\pi$ to assign fewer mass to the action $\mathbf{a}$.

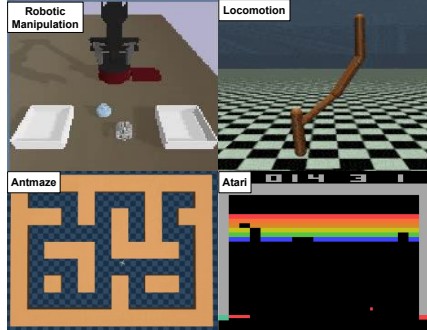

Figure 2: Our method outperforms offline RL methods based on distribution constraints on a diverse set of domains.

In Eq. 3 $\alpha$ is a constant, and hence the value function is modified by the ratio of probabilities to the same extent at all possible state-action pairs. Similarly, the $\alpha$ in Eq. 1 weights the two terms in the objective equally. Such design choices can be sub-optimal when the learnt policy should stay close to the behavior policy in some states, but not others. We next elaborate on this intuition both theoretically and experimentally.

## 3    WHY DISTRIBUTION CONSTRAINTS FAIL WITH HETEROSKEDASTIC DATA

Realistic sources of offline data are often heteroskedastic. Heteroskedasticity is where for certain regions of the state space, the observed behaviors assign the most probability mass to a few actions, while in other regions, the behaviors are more uniform. This is because datasets are typically generated by multiple policies, each with its own characteristics under a variety of different conditions. For instance, driving datasets are typically collected from multiple humans (Ettinger et al., 2021), and many robotic manipulation datasets are collected by multiple scripted policies (Mandlekar et al., 2021) or teleoperators (Ebert et al., 2021) that aim to solve a variety of tasks.

### 3.1    A DIDACTIC EXAMPLE

To understand why distributional constraints are insufficient when learning from heteroskedastic data, we present a didactic example. Motivated by the driving scenario in the introduction, we consider a maze navigation task shown in Fig. 1. The task is to navigate from the position labeled as "Start" to the position labeled as "Goal" using five actions at every possible state (L: ←, R: →, U: ↑, D: ↓, No: No Op), while making sure that the executed actions do not hit the walls of the grid.

**Heteroskedastic dataset.** In order to collect a heteroskedastic dataset, we consider a mixture of several behavior policies that attain a uniform occupancy over different states in the maze. However, the dataset action distributions differ significantly in different states. The induced action distribution is heavily biased to move towards the goal in the narrow hallways (e.g., the behavior policy moves upwards at state A)). In contrast, the action distribution is quite diverse in the wider rooms. In these rooms, the behavior policy often selects actions that do not immediately move the agent towards the goal (e.g., the behavior policy at state B), because doing so does not generally hit the walls as the rooms are wider, and hence the agent is not penalized. On the other hand, the agent must take utmost precaution to not hit the walls in the narrow hallways. More details are in Appendix A.

**Representative distribution constraints algorithms** such as AWR (Peng et al., 2019; Nair et al., 2020b) and CQL (Kumar et al., 2020) fail to perform the task, as shown in Figure 1. To ensure fair comparison, we tune each method to its best evaluation performance using online rollouts. The visualization in Figure 1 demonstrates that these two algorithms fail to learn reasonable policies because the learned policies match the random behavior of the dataset actions too closely in the wider rooms, and therefore are unable to make progress towards the Goal position. On the other hand, as shown in Appendix A, utilizing a lower amount of conservatism is unable to prevent the policies from hitting the walls in the narrow hallways. In contrast, the method we propose in this paper, indicated as "CQL (ReDS)", effectively traverses the maze and attains a success rate of 80%.

## 3.2 Characterizing the Challenges with Distributional Constraints

Having seen that distribution constraints can fail in certain scenarios, we now formally characterize when offline RL datasets is heteroskedastic, and why distribution constraints may be ineffective in such scenarios. Similar to how standard analyses utilize concentrability coefficient (Rashidinejad et al., 2021), which upper bounds the ratio of state-action visitation under a policy $d^\pi(\mathbf{s}, \mathbf{a})$ and the dataset distribution $\mu$, i.e., $\max_{\mathbf{s},\mathbf{a}} d^\pi(\mathbf{s}, \mathbf{a})/\mu(\mathbf{s}, \mathbf{a}) \leq C^\pi$, we introduce a new metric called *differential concentrability*, which measures dataset heteroskedasticity. All proofs are in Appendix B.

**Definition 3.1** (Differential concentrability.). *Given a divergence $D$ over the action space, the differential concentrability of a given policy $\pi$ with respect to the behavioral policy $\pi_\beta$ is given by:*

$$C_{diff}^\pi = \mathop{\mathbb{E}}_{\mathbf{s}_1, \mathbf{s}_2 \sim d^\pi} \left[ \left( \sqrt{\frac{D(\pi, \pi_\beta)(\mathbf{s}_1)}{\mu(\mathbf{s}_1)}} - \sqrt{\frac{D(\pi, \pi_\beta)(\mathbf{s}_2)}{\mu(\mathbf{s}_2)}} \right)^2 \right]. \tag{4}$$

Eq. 4 measures the variation in the divergence between a given policy $\pi(\mathbf{a}|\mathbf{s})$ and the behavior policy $\pi_\beta(\mathbf{a}|\mathbf{s})$ weighted inversely by the density of these states in the offline dataset (i.e., $\mu(\mathbf{s})$ in the denominator). For simplicity, let us revisit the navigation example from Section 3.1 and first consider a scenario where $\mu(\mathbf{s}) = \text{Unif}(\mathcal{S})$. If there are states where $\pi$ chooses actions that lie on the fringe of the data distribution (e.g., in the wider rooms), as well as states where the policy $\pi$ chooses actions at the mode of the data distribution (e.g., as in the narrow passages), then $C_{\text{diff}}^\pi$ would be large for the policy $\pi$ learned using distributional constraints. Crucially, $C_{\text{diff}}^\pi$ would be small even if the learned policy $\pi$ deviates significantly from the behavior policy $\pi_\beta$, such that $D(\pi, \pi_\beta)(\mathbf{s})$ is large, but $|D(\pi, \pi_\beta)(\mathbf{s}_1) - D(\pi, \pi_\beta)(\mathbf{s}_2)|$ is small, indicating the dataset is not heteroskedastic.

We will show in Section 5 that arbitrary policy checkpoints $\pi$ generally attain a low value of $C_{\text{diff}}^\pi$ on offline datasets from non-heteroskedastic datasets such as those D4RL (Fu et al., 2020). We now use differential concentrability to bound both the improvement and deprovement of $\pi$ w.r.t. $\pi_\beta$ for distribution constraint algorithms using the framework of safe policy improvement (Laroche et al., 2017; Kumar et al., 2020). We show that when $C_{\text{diff}}^\pi$ is large, then distribution constraints (Eq. 1) may not improve significantly over $\pi_\beta$, even for the best value for the weight $\alpha$:

**Theorem 3.1** (Informal; Limited policy improvement via distributional constraints.). *W.h.p. $\geq 1-\delta$, for any prescribed level of safety $\zeta$, the maximum possible policy improvement over choices of $\alpha$, $\max_\alpha \ [J(\pi_\alpha) - J(\pi_\beta)] \leq \zeta^+$, where $\zeta^+$ is given by:*

$$\zeta^+ := \max_\alpha \quad \frac{h^*(\alpha)}{(1-\gamma)^2} \quad s.t. \quad \frac{c_1 \sqrt{\log \frac{|\mathcal{S}||\mathcal{A}|}{\delta}}}{(1-\gamma)^2} \frac{\sqrt{C_{diff}^{\pi_\alpha}}}{|\mathcal{D}|} - \frac{\alpha}{1-\gamma} \mathbb{E}_{\mathbf{s} \sim \widehat{d}^{\pi_\alpha}} [D(\pi_\alpha, \pi_\beta)(\mathbf{s})] \leq \zeta, \quad (5)$$

*where $h^*$ is a monotonically decreasing function of $\alpha$, and $h(0) = \mathcal{O}(1)$.*

Theorem 3.1 quantifies the fundamental tradeoff with distribution constraints: to satisfy a given $\zeta$-safety constraint in problems with larger $C_{\text{diff}}^\pi$, we would need a larger $\alpha$. Since the maximum policy improvement $\zeta^+$ is upper bounded by $h^*(\alpha)$, the policy may not necessarily improve over the behavior policy if $\alpha$ is large. On the flip side, if we choose to fix the value of $\alpha$ to be small in hopes to attain more improvement in problems where $C_{\text{diff}}^\pi$ is high, we would end up compromising on the safety guarantee as $\zeta$ needs to be large for a small $\alpha$ and large $C_{\text{diff}}^\pi$. Thus, in this case, the policy may not improve over the behavior policy reliably. Note that larger $C_{\text{diff}}^\pi$ need not imply large $\mathbb{E}_{\mathbf{s} \sim \widehat{d}^\pi} [D(\pi, \pi_\beta)(\mathbf{s})]$ because the latter does not involve $\mu(\mathbf{s})$. $C_{\text{diff}}^\pi$ also measures the dispersion of $D(\pi, \pi_\beta)(\mathbf{s})$, while the latter performs a mean over states. In addition, Theorem 3.1 characterizes the *maximum possible* improvement with an *oracle* selection of $\alpha$, though is not feasible in practice. Thus, when $C_{\text{diff}}^\pi$ is large, distribution constraint algorithms could either not safely improve over $\pi_\beta$ or would attain only a limited, bounded improvement.

## 4 Support Constraints As Reweighted Distribution Constraints

Thus far, we have seen that distribution constraints can be ineffective with heteroskedastic datasets. If we can impose the distribution constraints such that the constraint strength can be modulated per state, then in principle, we can break the upper bound on performance shown in Theorem 3.1.

**Our key insight** is that by reweighting the action distribution in the dataset before utilizing a distribution constraint, we can obtain a method that enforces a per-state distribution constraint, which corresponds to an approximate *support* constraint. This will push down the values of actions that

(a) Support constraints is preferable to distribution constraints     (b) The strength of the constraints is modulated per state

Figure 3: **Comparison between support and distributional constraints: (a) Left:** CQL pushes down the Q-function under the policy $\pi$ (red) **(1)**, while pushing up the function under the behavior policy $\pi_\beta$. This means that the Q-values for bad actions can go up **(2)**. **(a) Right:** In contrast, ReDS re-weights the data distribution to push down the values of bad actions **(3)**, alleviating this shortcoming **(4)**. **(b) Left:** the two modes (5) and (6) of the behavior policy $\pi_\beta$ have similar Q values. The reweighted distribution (in red) pushes down the values of the two modes to similar extent. The constraint requiring the learned policy to stay *distributionally* close to the behavior policy is strong in this state (not an issue given the two modes have similar values). **(b) Right:** The reweighted distribution pushes down less (7) the mode having clearly higher Q-value. The distributional constraint is thus only weakly enforced in this case, allowing the learned policy to pick out the better mode.

are outside the behavior policy support, but otherwise not impose a severe penalty for in-support actions, thus enabling the policy to deviate from the behavior policy by different amounts at different states. Consider the generic distribution constraint shown in Eq. 1. Rather than having a distribution constraint between $\pi$ and $\pi_\beta$, if we can impose a constraint between $\pi$ and a *reweighted* version of $\pi_\beta$, where the reweighting is state-dependent, then we can obtain an approximate support constraint. Let the reweighted distribution be $\pi^{re}$. Intuitively, if $\pi(\cdot|\mathbf{s})$ is within the support of the $\pi_\beta(\cdot|\mathbf{s})$, then one can find a reweighting $\pi^{re}(\cdot|\mathbf{s})$ such that $D(\pi, \pi^{re})(\mathbf{s}) = 0$, whereas if $\pi(\cdot|\mathbf{s})$ is not within the support of $\pi^{re}(\cdot|\mathbf{s})$, then $D(\pi, \pi^{re})(\mathbf{s})$ still penalizes $\pi$ when $\pi$ chooses out-of-support actions, since no reweighting $\pi^{re}$ can put non-zero probability on out-of-support actions. This allows us to handle the failure mode from Section 3: at states with wide behavior policy, even with a large $\alpha$, $\pi$ is not anymore constrained to the behavior policy distribution, whereas at other "critical" states, where the behavior policy is narrow, a large enough $\alpha$ will constrain $\pi(\cdot|\mathbf{s})$ to stay close to $\pi_\beta(\cdot|\mathbf{s})$. We call this approach **Re**weighting **D**istribution constraints to **S**upport (ReDS).

### 4.1 Instantiating the Principle Behind ReDS

How can we derive a reweighting of the behavior policy to instantiate an approximate support constraint? One option is reweight $\pi_\beta$ into $\pi^{re}$, and enforce the constraints between $\pi$ and $\pi^{re}$ explicitly, by minimizing $D(\pi, \pi^{re})$, or implicitly, by pushing up the Q-values under $\pi^{re}$ instead of $\pi_\beta$, analogously to CQL. Both choices are problematic, because the error in estimating the behavior policy can propagate, leading to poor downstream RL performance, as demonstrated in prior work (Nair et al., 2020b; Ghasemipour et al., 2021). For the case of CQL, this issue might be especially severe if we push up the Q-values under $\pi^{re}$, because then the errors in estimating the behavior policy might lead to severe over-estimation.

Instead, we devise an alternative formulation for ReDS that re-weights the learned policy such that applying a distributional constraint on this re-weighting imposes a support constraint. In this formulation, we instead *push down* the Q-values under $\pi^{re}$. To still enable approximate support constraints between $\pi$ and $\pi_\beta$, we define $\pi^{re}$ as a mixture distribution of the learned policy $\pi$ and the reweighted behavior policy as shown below. We then demonstrate how $\pi^{re}$ can modify CQL to enable approximate support constraint-s while re-using all existing algorithmic components.

$$\pi^{re}(\cdot|\mathbf{s}) := \frac{1}{2}\pi(\cdot|\mathbf{s}) + \frac{1}{2}\left[\pi_\beta(\cdot|\mathbf{s}) \cdot g\left(\pi(\cdot|\mathbf{s})\right)\right], \qquad (6)$$

where $g(\cdot)$ is a monotonically decreasing function. As Fig. 3a depicts, the second term in Eq. 6 increases the probability of actions that are likely under the behavior policy, but are less likely under the learned policy (due to $g$ being a decreasing function). We will show in Lemma 4.1 that utilizing $\pi^{re}$ in CQL enforces an approximate support constraint on $\pi$. Utilizing a mixture of $\pi$ and a reweighted $\pi_\beta \cdot g$ is crucial, allowing for two important properties. Firstly, the inclusion of $\pi$ in the mixture ensures that the learned policy $\pi$ is penalized for deviating outside the support of the behavior policy, $\pi_\beta$. Second, the inclusion of the re-weighted behavior policy $\pi_\beta \cdot g(\mathbf{a}|\mathbf{s})$ allows $\pi$ to assign more density to good actions that are within the support of $\pi_\beta$, even if these actions have lower densities under $\pi_\beta$. This ensures that $\pi$ is constrained to lie within the support of $\pi_\beta$, but is

not constrained to the behavior policy distribution. Section 4.2 illustrates theoretically why pushing down the Q-values under Eq. 6 approximates a support constraint in terms of how it modifies the resulting Q-values. We must pick $g$ to devise a complete offline algorithm. A practical choice that reuses existing components in CQL is to subsume $\pi_\beta$ and $g$ into one distribution, denoted by $\rho$ (additional justification below). More concretely, $\pi^{re}$ now mixes $\pi$ and $\rho$ (the equal weight of 1/2 is an arbitrary choice and any non-zero weight on $\rho$ can be used):

$$\pi^{re}(\cdot|\mathbf{s}) := \frac{1}{2}\pi(\cdot|\mathbf{s}) + \frac{1}{2}\rho(\cdot|\mathbf{s}) \tag{7}$$

ReDS modifies the regularizer $\mathcal{R}(\theta)$ in CQL (Eq.3) to utilize the mixture distribution $\rho$ from Eq. 7, leading to the following modified regularizer:

$$\mathcal{R}(\theta;\rho) = \left(\frac{1}{2}\mathbb{E}_{\mathbf{s}\sim\mathcal{D},\mathbf{a}\sim\pi}[Q_\theta(\mathbf{s},\mathbf{a})] + \frac{1}{2}\mathbb{E}_{\mathbf{s}\sim\mathcal{D},\mathbf{a}\sim\rho}[Q_\theta(\mathbf{s},\mathbf{a})] - \mathbb{E}_{\mathbf{s},\mathbf{a}\sim\mathcal{D}}[Q_\theta(\mathbf{s},\mathbf{a})]\right) \tag{8}$$

We will show that Equation 8 imposes a support constraint in Lemma 4.1. Subsuming $\pi_\beta$ and $g$ into $\rho$ means the optimal $\rho$ should satisfy $\rho(\cdot|\mathbf{s}) = \pi_\beta(\cdot|\mathbf{s}) \cdot g(\pi(\cdot|\mathbf{s}))$. Under the MaxEnt RL framework (Levine, 2018), policy probabilities are the exponentiated advantages: $\pi(\mathbf{a}|\mathbf{s}) = \exp(A(\mathbf{s},\mathbf{a})) = \exp(Q(\mathbf{s},\mathbf{a}) - V(\mathbf{s}))$. As such, we can utilize the advantage function (derived from Q-function) directly to train $\rho$, instead of using the policy. That is, $\rho(\cdot|\mathbf{s}) = \pi_\beta(\cdot|\mathbf{s}) \cdot g(\exp(A(\mathbf{s},\mathbf{a})))$. As shown in prior work on reward-weighted and advantage-weighted policy learning (e.g., AWR (Peng et al., 2019)), we can solve for this distribution using a weighted maximum log-likelihood objective. We choose $g(x) = 1/x$, such that $g(\exp(A(\mathbf{s},\mathbf{a}))) = \exp(-A(\mathbf{s},\mathbf{a}))$ (which is a decreasing function). As is common with such weighted regression methods (Peng et al., 2019), we add a temperature $\tau$ into the exponent, yielding the following objective for $\rho$:

$$\rho_\psi(\cdot|\mathbf{s}) = \arg\max_{\rho_\psi} \ \mathbb{E}_{\mathbf{s}\sim\mathcal{D},\mathbf{a}\sim\pi_\beta(\cdot|\mathbf{s})}[\log\rho_\psi(\mathbf{a}|\mathbf{s}) \cdot \exp(-A_\theta(\mathbf{s},\mathbf{a})/\tau)]. \tag{9}$$

The crucial difference between this objective and standard advantage-weighted updates is the difference of the sign. While algorithms such as AWR aim to find an action that attains a high advantage while being close to the behavior policy, and hence, uses a positive advantage, we utilize the *negative* advantage to mine for poor actions that are still quite likely under the behavior policy. The final objective for the Q-function combines the regularizer in Eq. 8 with a standard TD objective:

$$\min_\theta \quad J_Q(\theta) = \mathcal{R}(\theta;\rho) + \frac{1}{2}\mathbb{E}_{\mathbf{s},\mathbf{a},\mathbf{s}'\sim\mathcal{D}}\left[\left(Q_\theta(\mathbf{s},\mathbf{a}) - \mathcal{B}^\pi\bar{Q}(\mathbf{s},\mathbf{a})\right)^2\right] \tag{10}$$

We name the new algorithm CQL (ReDS), illustrated in Algorithm 1. Description using Python-syntax is in Appendix C.1. CQL (ReDS) first performs one gradient descent step on $\rho_\psi$ using Eq. 9, then performs one gradient step on the Q-function using the objective in Eq. 10, then performs one gradient descent step on the policy $\pi_\phi$, and repeats.

---

**Algorithm 1** CQL (ReDS) pseudo-code

1: Initialize Q-function, $Q_\theta$, a policy, $\pi_\phi$, and distribution $\rho_\psi$
2: **for** step $t$ in $\{1, \ldots, N\}$ **do**
3:    Update $\rho_\psi$ with the objective in Eq. 9:
    $\psi_t := \psi_{t-1} + \eta_\rho \nabla_{\psi} \mathbb{E}_{\mathbf{s},\mathbf{a}\sim\mathcal{D}}[\log\rho_\psi(\mathbf{a}|\mathbf{s}) \cdot \exp(-A_\theta(\mathbf{s},\mathbf{a})/\tau)]$
4:    Train the Q-function using $J_Q(\theta)$ in Eq. 10:
    $\theta_t := \theta_{t-1} - \eta_Q \nabla_\theta J_Q(\theta)$
5:    Improve policy $\pi_\phi$ with SAC-style update:
    $\phi_t := \phi_{t-1} + \eta_\pi \nabla_\phi \mathbb{E}_{\mathbf{s}\sim\mathcal{D},\mathbf{a}\sim\pi_\phi(\cdot|\mathbf{s})}[Q_\theta(\mathbf{s},\mathbf{a}) - \log\pi_\phi(\mathbf{a}|\mathbf{s})]$

---

### 4.2 THEORETICAL ANALYSES OF CQL (REDS)

Next, we analyze CQL (ReDS), showing how learning using the regularizer in Eq.(7) modifies the Q-values, providing justification behind our choice of the distribution $\rho$ in the previous section.

**Lemma 4.1** (Per-state modification of Q-values.). *Let $g$ represents $g(\tau \cdot \pi(\cdot|\mathbf{s}))$. In tabular setting, the Q-function obtained after one TD-learning iteration using the objective in Eq. 10 is:*

$$Q_\theta(\mathbf{s},\mathbf{a}) := \mathcal{B}^\pi\bar{Q}(\mathbf{s},\mathbf{a}) - \alpha\frac{\pi(\mathbf{a}|\mathbf{s}) + \pi_\beta(\mathbf{a}|\mathbf{s})g - 2\pi_\beta(\mathbf{a}|\mathbf{s})}{2\pi_\beta(\mathbf{a}|\mathbf{s})} \tag{11}$$

*where $\mathcal{B}^\pi\bar{Q}(\mathbf{s},\mathbf{a})$ is the Bellman backup operator applied to a delayed target Q-network.*

Eq. 39 illustrates why the modified regularizer in Eq. 8 leads to a "soft" support constraint whose strength is modulated per-state. Since $g$ is a monotonically decreasing function of $\pi$, for state-action pairs where $\pi(\mathbf{a}|\mathbf{s})$ has high values, $g$ is low and therefore the $Q$ values for such state-action pairs

are underestimated less. Vice versa, for state-action pairs where $\pi(\mathbf{a}|\mathbf{s})$ attains low values, $g$ is high to counter-acts the low $\pi(\mathbf{a}|\mathbf{s})$ values. Also, since $\pi_\beta(\mathbf{a}|\mathbf{s})$ appears in the denominator, for out-of-support actions, where $\pi_\beta(\mathbf{a}|\mathbf{s}) = 0$, $\pi(\mathbf{a}|\mathbf{s})$ must also assign $0$ probability to the actions for the Q values to be well defined. We can further formalize the objective for ReDS as follows:

**Lemma 4.2** (CQL (ReDS) objective.). *Assume that for all policies $\pi \in \Pi, \forall(\mathbf{s}, \mathbf{a}), \pi(\mathbf{a}|\mathbf{s}) > 0$. Then, CQL (ReDS) solves the following optimization problem:*

$$\max_{\pi \in \Pi} \widehat{J}(\pi) - \frac{\alpha}{2(1-\gamma)} \mathbb{E}_{\mathbf{s} \sim \widehat{d}^\pi} \left[ D(\pi, \pi_\beta)(\mathbf{s}) + \cdot \mathbb{E}_{\mathbf{a} \sim \pi(\cdot|\mathbf{s})} \left[ g\left(\tau \cdot \pi(\mathbf{a}|\mathbf{s})\right) \mathbb{I}\left\{\pi_\beta(\mathbf{a}|\mathbf{s}) > 0\right\} \right] \right]. \quad (12)$$

$\widehat{J}(\pi)$ corresponds to the empirical return of the learned policy, i.e., the return of the policy under the learned Q-function. The objective in Lemma 4.2 can be intuitively interpreted as follows. The first term, $D(\pi, \pi_\beta)(\mathbf{s})$, is a standard distributional constraint, also present in naïve CQL, and it aims to penalize the learned policy $\pi$ if it deviates too far away from $\pi_\beta$. ReDS adds an additional second term that effectively encourages $\pi$ to be "sharp" within the support of the behavior policy, enabling it to put its mass on actions that lead to a high $\widehat{J}(\pi)$. This second term allows us control the strength of the distributional constraint per state: at states where the support of the policy is narrow, i.e., the volume of actions such that $\pi_\beta(\mathbf{a}|\mathbf{s}) \geq \varepsilon$ is small, the penalty in Equation 12 reverts to a standard distributional constraint by penalizing divergence from the behavioral policy via $D(\pi, \pi_\beta)(\mathbf{s})$ as the second term is non-existent. At states where the policy $\pi_\beta$ is broad, the second term counteracts the effect of the distributional constraint within the support of the behavior policy, by $\pi$ to concentrate its density on only good actions within the support of $\pi_\beta$ with the same multiplier $\alpha$. Thus even when we need to set $\alpha$ to be large in order to stay close to $\pi_\beta(\cdot|\mathbf{s})$ at certain states (e.g., in the narrow hallways in the didactic example in Sec. 3.1), $D(\pi, \pi_\beta)(\mathbf{s})$ is not heavily constrained at other states.

## 5 EXPERIMENTAL EVALUATION

The goal of our experiments is to understand how CQL (ReDS) compares to distributional constraint methods when learning from heteroskedastic offline datasets. In order to perform our experiments, we construct new heteroskedastic datasets that pose challenges representative of what we would expect to see in real-world problems. We first introduce tasks and heteroskedastic datasets that we evaluate on, and then present our results compared to prior state-of-the-art methods. We also evaluate ReDS on some of the standard D4RL (Fu et al., 2020) datasets which are not heteroskedastic in and find that the addition of ReDS, as expected, does not help, nor does it hurt on those tasks.

### 5.1 COMPARISON ON D4RL BENCHMARK

Our motivation for studying heteroskedastic datasets is that heteroskedasticity likely exists in real-world domains such as driving, manipulation where datasets are collected by multiple policies that agree and disagree at different states. While standard benchmarks (D4RL (Fu et al., 2020) and RLUnplugged (Gulcehre et al., 2020)) include offline datasets generated by mixture policies, e.g. the "medium-expert" generated by two policies with different performance, these policies are themselves trained via RL methods such as SAC, that constrain the entropy of the action distribution at each state to be uniform. To measure their heteroskedasticity, we utilize a tractable approximation to $C_{\text{diff}}^\pi$ used in our theoretical analysis: the standard deviation in the value of $D(\pi, \pi_\beta)(\mathbf{s})$ across states in the dataset, using a fixed policy $\pi$ obtained by running CQL. We did not use $C_{\text{diff}}^\pi$ directly, as it requires estimating the state density of the data distribution, which is challenging in continuous spaces. Observe in Table 4a that the standard deviation is relatively low for the D4RL antmaze datasets. This corroborates our intuition that D4RL datasets are significantly less heteroskedastic.

### 5.2 COMPARISONS ON CONSTRUCTED HETEROSKEDASTIC DATASETS

**Heteroskedastic datasets.** To stress-test our method and prior distributional constraint approaches, we collect new datasets for the medium and large mazes used in the antmaze navigation tasks from D4RL: `noisy` datasets, where the behavior policy action variance differs in different regions of the maze, representative of user variability in navigation, and `biased` datasets, where the behavior policy admits a systematic bias towards certain behaviors in different regions of the maze, representative of bias towards certain routes in navigation problems. Table 4a illustrates that these datasets are significantly more heteroskedastic compared to the D4RL datasets.

Using these more heteroskedastic datasets, we compare CQL (ReDS) with CQL and IQL (Kostrikov et al., 2021a), which are recent popular methods, and two prior methods, BEAR (Kumar et al., 2019) and EDAC (An et al., 2021), that also enforce support constraints. For each algorithm, including ours, we utilize hyperparameters directly from the counterpart tasks in D4RL. Due to lack of an

| Dataset | std | max |
|---|---|---|
| noisy (Ours) | **18** | **253** |
| biased (Ours) | **9** | **31** |
| diverse (D4RL) | 2 | 11 |
| play (D4RL) | 2 | 13 |

(a) the new antmaze datasets (Ours) are significantly more heteroskedastic compared to the standard D4RL datasets. We measure heteroskedasticity using the std and max of $D(\pi, \pi_\beta)(\mathbf{s})$ across states in the offline dataset.

| Task & Dataset | EDAC | BEAR | CQL | IQL | Ours |
|---|---|---|---|---|---|
| medium-noisy | 0 | 0 | 55 | 44 | **73** |
| medium-biased | 0 | 0 | **73** | 48 | **74** |
| large-noisy | 0 | 0 | 42 | 39 | **53** |
| large-biased | 0 | 0 | **50** | 41 | 45 |

(b) CQL (ReDS) (Ours) outperforms prior offline RL methods including state-of-the-art methods (IQL) , and prior support constraint methods(BEAR, EDAC) on three out of four scenarios when learning from heteroskedastic datasets in the antmaze task. The improvement over prior methods is larger when learning from the `noisy` datasets, which are more heteroskedactic, as shown in Table 4a, compared to the `biased` datasets.

Table 1: Our method out-performs prior methods in the antmaze navigation task when learning from heteroskedastic datasets, indicating the importance of support constraints relative to distributional constraints.

Table 2: Performance comparison with recent offline RL algorithms on the D4RL benchmark.

| Dataset | BC | 10%BC | DT | AWAC | Onestep RL | TD3+BC | COMBO | CQL | IQL | (Ours) |
|---|---|---|---|---|---|---|---|---|---|---|
| halfcheetah-medium-replay | 36.6 | 40.6 | 36.6 | 40.5 | 38.1 | 44.6 | 55.1 | 45.5 | 44.2 | 52.3 |
| hopper-medium-replay | 18.1 | 75.9 | 82.7 | 37.2 | 97.5 | 60.9 | 89.5 | 95.0 | 94.7 | 101.5 |
| walker2d-medium-replay | 26.0 | 62.5 | 66.6 | 27.0 | 49.5 | 81.8 | 56.0 | 77.2 | 73.9 | 85.0 |
| halfcheetah-medium-expert | 55.2 | 92.9 | 86.8 | 42.8 | 93.4 | 90.7 | 90.0 | 91.6 | 86.7 | 89.5 |
| hopper-medium-expert | 52.5 | 110.9 | 107.6 | 55.8 | 103.3 | 98.0 | 111.1 | 105.4 | 91.5 | 110.0 |
| walker2d-medium-expert | 107.5 | 109.0 | 108.1 | 74.5 | 113.0 | 110.1 | 103.3 | 108.8 | 109.6 | 112.0 |
| locomotion total | 295.9 | 491.8 | 488.4 | 277.8 | 494.8 | 486.1 | 505 | 523.5 | 500.6 | 550.3 |

effective method for offline policy selection (see Fu et al. (2021)), we utilize oracle checkpoint selection for every method. We compute the mean and standard deviation of the performance across 3 seeds. Table 4b illustrates that the largest gap between CQL (ReDS) and prior methods appears on the `noisy` datasets, which are particularly more heteroskedastic as measured in Table 4a.

We also compare CQL (ReDS) with recent offline RL algorithms on D4RL, including DT (Chen et al., 2021), AWAC (Nair et al., 2020a), onestep RL (Brandfonbrener et al., 2021), TD3+BC (Fujimoto & Gu, 2021b) and COMBO (Yu et al., 2021). Table 2 shows that CQL (ReDS) obtains similar performance as existing distributional constraint methods and outperform BC-based baselines. This is expected given that the D4RL datasets exhibit significantly smaller degree of heteroscedasticity, as previously explained. Also, a large fraction of the datasets is trajectories with high returns. BC using the top 10% trajectories with the highest episode returns already has strong performance (10%BC).

The previous results compares CQL (ReDS) to baselines in tasks where the MDP states are low-dimensional vectors. This section compares CQL (ReDS) to baselines in vision-based tasks.

**Visual robotic manipulation.** We consider two manipulation tasks. In the "Pick & Place" task, the algorithm controls a WidowX robot to grasp an object and place it into a tray located at a test location, directly from raw 128×128×3 images and sparse 0/1 reward signal. The dataset consists of behavior from suboptimal grasping and placing policies, and the positions of the tray in the offline dataset very rarely match the target test location. The placing policies exhibit significant variability, implying that these datasets are heteroskedastic under our definition. We also consider "Bin Sort" task, where a WidowX robot is controlled to sort two objects into two separate bins. Here heteroskedacity is introduced by when sorting the objects into the desirable bins. Similar to the Pick & Place domain, the placing policy exhibits significant variability, and shows an the object being placed in the incorrect bin (e.g., recyclable trash thrown into the non-recyclable bin). However, the grasping policy is narrow, grasping the object with low variability. More details in Appendix D.

Table 3 illustrates the results on these tasks. We utilize oracle policy selection analogous to the antmaze experiments from Table 4b. Table 3 shows that CQL (ReDS) outperforms CQL attaining a success rate of about 15.1% for the visual pick and place task, whereas CQL only attains 6.5% suc-

Table 3: **CQL (ReDS) vs CQL** on the manipulation tasks. CQL (ReDS) outperforms CQL significantly when learning from more heteroskedastic datasets, as measured by our estimates of $C^\pi_{\text{diff}}$: the standard-deviation and the maximum of $D(\pi, \pi_\beta)(\mathbf{s})$ across states.

| Task | CQL | CQL (ReDS) | std $D(\pi, \pi_\beta)(\mathbf{s})$ | max $D(\pi, \pi_\beta)(\mathbf{s})$ |
|---|---|---|---|---|
| Pick & Place | $6.5 \pm 0.4$ | $\mathbf{15.1 \pm 0.4}$ | **48.7** | **307.4** |
| Bin Sort (Easy) | $\mathbf{31.2 \pm 0.3}$ | $31.4 \pm 0.3$ | 7.9 | 81.6 |
| Bin Sort (Hard) | $6.1 \pm 0.2$ | $\mathbf{23.1 \pm 0.7}$ | **59.6** | **988.3** |

cess. While these performance numbers might appear low in an absolute sense, note that both CQL and ReDS do improve over the behavior policy, which only attains a success rate of 4%. Thus offline RL does work on this task, and utilizing ReDS in conjunction with the standard distributional constraint in CQL does result in a boost in performance with this heteroskedastic dataset. For the "Bin Sorting", our method outperforms CQL by **3.5x** when learning from more heteroskedastic datasets. This indicates that the effectiveness of our method in settings with higher heteroskedasticity.

**Atari games.** We collect data on 10 Atari games from multiple policies that behave differently at certain states, while having similar actions otherwise. We consider a case of **two** such policies, and a harder scenario of **five**. We evaluate the performance of CQL (ReDS) on the Atari games using the evaluation metrics from prior works (Agarwal et al., 2020; Kumar et al., 2021), denoted IQM.

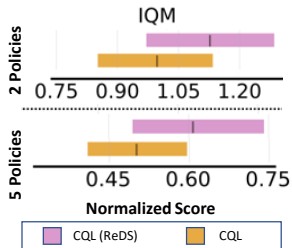

Figure 4: **Performance of CQL (ReDS) vs CQL** on the IQM normalized score and the mean normalized score over ten Atari games, for the case of **two** (top) and **five** (bottom) policies.

Figure 4 shows that in both testing scenarios: with the mixture of two policies (top figure) and the mixture of five policies (bottom figure), CQL (ReDS) outperforms CQL in aggregate.

**To summarize**, our results indicate that incorporating CQL (ReDS) outperform common distributional constraints algorithms when learning from heteroskedastic datasets.

## 6  RELATED WORK

Offline Q-learning methods utilize mechanisms to prevent backing up unseen actions (Levine et al., 2020), by applying an explicit behavior constraint that forces the learned policy to be "close" to the behavior policy (Jaques et al., 2019; Wu et al., 2019; Peng et al., 2019; Siegel et al., 2020; Wu et al., 2019; Kumar et al., 2019; Kostrikov et al., 2021b;a; Wang et al., 2020; Fujimoto & Gu, 2021a), or by learning a conservative value function (Kumar et al., 2020; Xie et al., 2021; Nachum et al., 2019; Yu et al., 2021; 2020; Rezaeifar et al., 2021; Jin et al., 2020; Wu et al., 2019). Most of these offline RL methods utilize a distributional constraint, explicit (e.g., TD3+BC (Fujimoto & Gu, 2021a)) or implicit (e.g., CQL (Kumar et al., 2020)), and our empirical analysis of representative algorithms from either family indicates that these methods struggle with heteroskedastic data, especially those methods that use an explicit constraint.

Some prior works have also made a case for utilizing support constraints instead of distribution constraints, often via didactic examples (Kumar et al., 2019; Kumar, 2019; Levine et al., 2020), and devised algorithms that impose support constraints in theory, by utilizing the maximum mean discrepancy metric (Kumar et al., 2019) or an asymmetric f-divergences (Wu et al., 2019) for the policy constraint (Wu et al., 2019). Empirical results on D4RL (Fu et al., 2020) and the analysis by Wu et al. (2019) suggest that support constraints are not needed, as strong distribution constraint algorithms often have strong performance. As we discussed in Sections 3.2 (Theorem 3.1 indicates that this distributional constraints may not fail when $C_{\text{diff}}^{\pi}$ is small, *provided these algorithms are well-tuned*.) and 4, these benchmark datasets are not heteroskedastic, as they are collected from policies that are equally wide at all states and centered on good actions (e.g., Antmaze domains in (Fu et al., 2020), control suite tasks in Gulcehre et al. (2020)) and hence, do not need to modulate the distributional constraint strength. To benchmark with heteroskedastic data, we developed some novel tasks which may be of independent interest beyond this work.

## 7  DISCUSSION

In this work, we studied the behavior of distributional constraint offline RL algorithms when learning from heteroskedastic datasets, a property we are likely encounter in the real world. Naïve distributional constraint algorithms can be highly ineffective in such settings both in theory and practice, as they fail to modulate the constraint strength per-state. We propose ReDS, a method to convert distributional constraints into support-based constraints via reweighting, and validate it in CQL. A limitation of ReDS is that it requires estimating the distribution $\rho_{\psi}$ to enforce a support constraint, which brings about its some additional compute overhead. Devising approaches for enforcing support constraints that do not require extra machinery is an interesting direction for future work. Understanding if support constraints are less sensitive to hyperparameters or are more amenable to cross-validation procedures is also interesting. Extending ReDS to other Q-learning based, and even model-based offline RL algorithms is also an interesting direction for future work.

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

# Appendices

## A   DETAILS OF THE DIDACTIC NAVIGATION EXAMPLE FROM SECTION 3.1

**AWR Visitations**

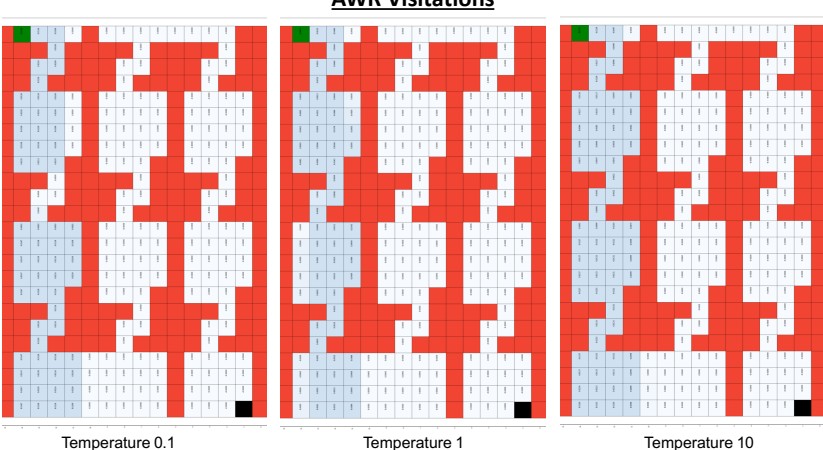

Figure 5: **Results of running AWR on the gridworld maze for a variety of temperature values** annotated with state-occupancy values. Observe that AWR is unable to reach the goal across a wide range of temperatures for the AWR hyperparameter. Even though for some of these hyperparameters, such as $\tau = 0.1$, it is able to traverse quickly into the third region, it does spend a larger fraction of its state visitation in the narrow hallways in this case (e.g., the final narrow hallway it is able to reach to) indicating that it gets stuck. Increasing the temperature to $\tau = 1.0$ does not solve it either, since now it does not even reach this hallway with as high of an occupancy.

**CQL Visitations**

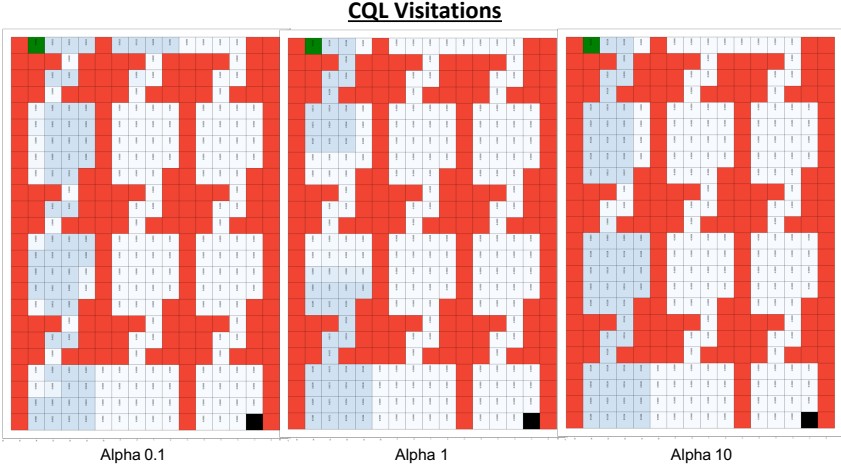

Figure 6: **Results of running CQL on the gridworld maze for a variety of temperature values.** Observe that CQL is unable to reach the goal and gets stuck for all the $\alpha$ values we studied.

In this section, we present some details regarding the navigation example we considered in Section 3.1. To create this example, we modified the gridworld code from Fu et al. (2019) (code taken from: https://github.com/justinjfu/diagnosing_qlearning) to create the corresponding gridworld maze. We utilize a $24 \times 16$ gridworld as shown in the figures below, larger than the $8 \times 8$ or $16 \times 16$ gridworlds studied in this repository in the past. We utilize a smooth representation of the observation space. This is constructed by first sampling random Gaussian feature vectors from $\mathbb{R}^{50}$ for each grid cell (each grid cell is a state). Smoothing of these features vectors is then done locally, following the protocol for "grid-*-smoothobs" in Fu et al. (2020). This observation type presents a challenge for Q-learning algorithms that may often generalize incorrectly. This is because aliasing occurs with the predictions of nearby states that exhibit very different dynamics

but not so different observation vectors. The agents gets a sparse binary 0/+1 reward: +1 is attained only when the agent reaches the goal (marked in black), starting from the start location (marked in green).

The behavior policy in each part of the maze is based on a mixture of different policies. In the wider rooms of the maze, one of the policies is a uniform policy, that uniformly chooses every action at a state. The second policy is a biased policy, that drives the agent away from the goal. In the narrow hallways, the behavior policy deterministically drives the agent towards the goal. For wide rooms, in contrast, a bias exists for actions taken in the direction away from the goal. This bias was set to be 0.8. This means that for rooms where you need to exit the wide passage by going down, the action of going up was selected 80% of the time by the behavior policy and each of the other action was randomly sampled with 20% probability. The opposite decision was taken for rooms where the goal was towards the top of the wide passage where 80% of the time the down action was selected.

**Result visualizations.** We now present some visualizations of the policies learned by various methods: AWR, CQL and CQL (ReDS). Since the reward values are binary and sparse, return curves for AWR and CQL are not as informative, since they attain a 0 return in any episode, even if they make some progress. Therefore, we present the results in the form of state-visitation density plots under rollouts from the learned policy. Observe in Figures 5 and 6 that neither AWR or CQL are able to actually successfully traverse the maze, and get stuck in it. Note that $\tau$ and $\alpha$ control the strength of the distributional constraint in the methods AWR and CQL respsectively. Varying the temperature hyperparameter $\tau$ for AWR and $\alpha$ for CQL does modify the density in the narrow hallways, but utilizing too small of a parameter leads the agent to more frequent crashes in the narrow hallways. This makes the agent spend a higher fraction of its visitation in one such narrow region, whereas a higher $\tau$ or $\alpha$ does not even reach the third narrow hallway with a high-enough visitation.

**REDS State Visitations**

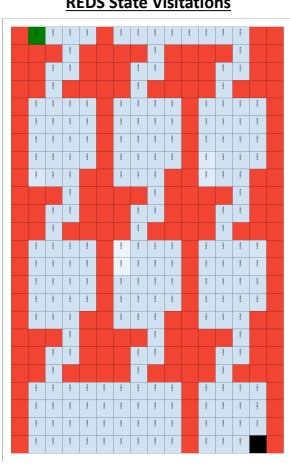

Figure 7: **Results of running CQL (ReDS) on the gridworld.** Note that CQL (ReDS) does actually succeed at solving the task.

On the other hand, the method we propose in this paper, ReDS, when combined with CQL is able to successfully traverse this maze, as shown in Figure 7.

**Analysis.** These experiments are consistent with our expectations. When distributional constraints are faced with highly heteroskedastic action distributions at subsequent states, they failed to perform good actions at these consecutive states. A strong distributional constraint leads to the agent being too close to the behavior policy, whereas a weaker constraint fails to identify good actions at states where the behavior policy is narrow.

## A.1 WHY AND WHEN DO DISTRIBUTIONAL CONSTRAINTS FAIL IN SCENARIOS WITH HETEROSKEDASTIC DATA?

In this section, we shall discuss why distributional constraints are especially worse than support constraints in scenarios with heteroskedastic data. Let's first consider a simple single-state bandit problem. In this simple one-state problem, we must find a single optimal action by using offline

data. At first, it might appear that when faced with a wide behavior policy, distributional constraints would clearly fail since they would not deviate far away from the behavior policy. However, note that by carefully choosing the strength of the distributional constraint, we can, in principle, control the strength of the distributional constraint quite effectively. To see this concretely, consider applying CQL (Equation 2) to a single-state bandit problem. The Q-values on actions not observed in the training dataset will be clearly pushed down to $-\infty$. In addition, choosing a non-zero $\alpha$ will allow us to precisely control how close the action taken by the learned policy is to the best in-support action vs how close the learned policy is to the behavior policy. Therefore, for an optimally chosen value of $\alpha$, distributional constraint methods such as CQL should already work well in the single-state setting.

However, the precise challenge with distributional constraints arises in the multi-state setting, where finding a single value of $\alpha$ that can work well at all states might be can be exceedingly challenging in practice. This is the setting with heteroskedastic data that our paper considers, including in our didactic example above. In such a setting, a strong distributional constraint is able to take a desirable action when the behavior policy is narrow (for example in the narrow room). However, the action distribution has wider support in the wider room and the strong distributional constraint would not take a desirable action in this setting due to the high entropy of the behavioral distribution in this setting. In contrast, with a weaker distributional constraint, a bad out-of-distribution action may be taken where the behavior policy is narrow which can lead to instability in the policy. Note that these challenges are not, however, present in the single-state scenario. Our method CQL (ReDS) can tackle the problem in this setting by modulating the strength of the constraint per state. This allows the agent to stay close to the behavior distribution in the narrow room as well as learning to output the most desirable in-support action in the wider room.

### A.2  HETEROSKEDASTIC DATA VS EXPLORATORY DATA

There is a distinction between exploratory data (Yarats et al., 2022) and heteroskedasticity. In the context of offline RL, exploratory data typically refers to being able to cover **all** possible state-action pairs, as uniformly as possible. On the other hand, heteroskedasticity is used to refer to **disproportionate** coverage: when the action distribution in the offline data is more uniform at certain states, but very narrow at others. Thus, a heteroskedastic dataset does not have high coverage in the sense of observing all possible state-action pairs.

Mathematically, while the notion of coverage is typically quantified by the concentrability coefficient which denotes the **worst case** density ratio between the distribution of the training dataset and the distribution of the learned policy, as shown below:

$$C^{\pi} = \max_{\mathbf{s},\mathbf{a}} \; \frac{d^{\pi}(\mathbf{s},\mathbf{a})}{\mu(\mathbf{s},\mathbf{a})},$$

where $\mu(\mathbf{s},\mathbf{a})$ denotes the state-action distribution induced by the dataset, while $d^{\pi}(\mathbf{s},\mathbf{a})$ denotes the state-action visitation distribution of a policy $\pi$. $\max_{\pi} C^{\pi}$ is generally expected to be smaller when $\mu(\mathbf{s},\mathbf{a})$ is more uniform. In contrast, we quantify the notion of heteroskedasticity by the variation in density ratios (in this case, action density ratios, $\frac{\pi(\mathbf{a}|\mathbf{s})}{\pi_{\beta}(\mathbf{a}|\mathbf{s})}$) across the state space in Equation 4 (reproduced below for convenience).

$$C^{\pi}_{\mathrm{diff}} = \mathbb{E}_{\mathbf{s}_1,\mathbf{s}_2 \sim d^{\pi}} \left[ \left( \sqrt{\frac{D(\pi,\pi_{\beta})(\mathbf{s}_1)}{\mu(\mathbf{s}_1)}} - \sqrt{\frac{D(\pi,\pi_{\beta})(\mathbf{s}_2)}{\mu(\mathbf{s}_2)}} \right)^2 \right].$$

Even if the dataset has high coverage (i.e., it has a low concentrability coefficient, $C^{\pi}$), it can attain a higher $C^{\pi}_{\mathrm{diff}}$, because of higher variation in the density ratios across state spaces.

**Out of all methods considered in Yarats et al. (2022)**, we note that the methods classified as "Data" (APT, ProtoRL) attempt to maximize state coverage, the methods classified as "Knowledge" (ICM, Disagreement, RND) attempt to find surprising state-action pairs and visit them, and the methods classified as "Competence" (SMM, DIAYN, APS) that maximize the mutual information between the state distribution attained by the learned policy and the learned skills also implicitly optimize for state coverage. In effect, all of these methods aim to visit all state-action pairs (or all states), which is akin to the standard definition of concentrability, whereas heteroskedastic data corresponds distinctly to disproportionate action coverage at different states.

## B PROOFS

In this appendix, we will provide proofs for the various theoretical results in the main paper: Theorem 3.1. We will first discuss some preliminaries and notation, then present the proof for Theorem 3.1, and finally the remaining results.

### B.1 NOTATION AND PRELIMINARIES

Let $\pi_\beta(\mathbf{a}|\mathbf{s})$ denote the behavior policy. Note that the dataset, $\mathcal{D}_i$ is generated from the marginal state-action distribution of $\pi_\beta$, i.e., $\mathcal{D} \sim d^{\pi_\beta}(\mathbf{s})\pi_\beta(\mathbf{a}|\mathbf{s})$. Define $\widehat{d^\pi}$ as the state marginal distribution introduced by $\pi$ under the empirical MDP defined by the transitions in the dataset. Let $D_{\mathrm{CQL}}(p, q)$ denote the following distance between two distributions $p(\mathbf{x})$ and $q(\mathbf{x})$ with equal support $\mathcal{X}$:

$$D_{\mathrm{CQL}}(p, q) := \sum_{\mathbf{x} \in \mathcal{X}} p(\mathbf{x}) \left( \frac{p(\mathbf{x})}{q(\mathbf{x})} - 1 \right).$$

We drop the subsrcipt "CQL" from $D_{\mathrm{CQL}}$ for clarity. (Kumar et al., 2020) showed that when optimizing the generic distributional constraint algorithm shown in Equation 1, the resulting policy $\pi^*$ attains a high probability safe-policy improvement guarantee, i.e., $J(\pi^*) \geq J(\pi_\beta) - \zeta$, where $\zeta$ is:

$$\zeta = \mathcal{O}\left( \frac{1}{(1-\gamma)^2} \right) \mathbb{E}_{\mathbf{s} \sim \widehat{d}^{\pi^*}} \left[ \sqrt{\frac{D(\pi^*, \pi_\beta)(\mathbf{s}) + 1}{|\mathcal{D}(\mathbf{s})|}} \right] + \frac{\alpha}{1-\gamma} \mathbb{E}_{\mathbf{s} \sim \widehat{d}^\pi} [D(\pi^*, \pi_\beta)(\mathbf{s})]. \quad (13)$$

We can further express $|\mathcal{D}(\mathbf{s})| = |\mathcal{D}||\mu(\mathbf{s})|$. The first term in Equation 13 corresponds to the decrease in performance due to sampling error and this term is high when the learned policy $\pi^*$ visits low density states under the dataset distribution (i.e., $\mu(\mathbf{s})$ is small) and when the divergence from the behavior policy $\pi_\beta$ is higher under these states. We will use this safe policy improvement guarantee in our proofs.

### B.2 FORMAL RESTATEMENT OF THEOREM 3.1

**Theorem B.1** (Formal version of Theorem 3.1). *For any prescribed level of safety $\zeta$, i.e., the learned policy satisfies $J(\pi) - J(\pi_\beta) \geq -\zeta$, the maximum possible policy improvement over choices of $\alpha$:*

$$\max_\alpha \left[ J(\pi_\alpha) - J(\pi_\beta) \right] \leq \zeta^+,$$

*where $\zeta^+$ is given by:*

$$\zeta^+ := \max_\alpha \quad h^*(\alpha) \cdot \frac{1}{(1-\gamma)^2} \quad s.t. \quad \frac{c_1 \sqrt{\log \frac{|\mathcal{S}||\mathcal{A}|}{\delta}}}{(1-\gamma)^2} \frac{\sqrt{C_{\mathit{diff}}^{\pi_\alpha}}}{|\mathcal{D}|} - \zeta \leq \alpha \cdot \underbrace{\left( \frac{\mathbb{E}_{\mathbf{s} \sim \widehat{d}^{\pi_\alpha}} [D(\pi_\alpha, \pi_\beta)(\mathbf{s})]}{1-\gamma} \right)}_{:=g(\alpha)}$$

$$(14)$$

*where $h^*$ is a monotonically decreasing function of $\alpha$, and $h(0) = const.$*

**Intuition behind the theorem:** We will show in the proof of this theorem that $g(\alpha)$ is a monotonically increasing function of $\alpha$ and $g(0) = 0$. This means that if the value of $C_{\mathrm{diff}}^\pi$ is large for all policies $\pi$, the value of $\alpha = g^{-1}\left( \sqrt{C_{\mathrm{diff}}^\pi}c' - \zeta \right)$ (where $c'$ is the constant and subsumes the term containing $\delta$) is also large. Further, note that $h^*(\alpha)$ is decreasing in $\alpha$. This means that the larger the value of $\alpha$, the smaller the value of $h^*(\alpha)$, and hence, smaller the value of $\zeta^+$. This means that as $C_{\mathrm{diff}}^\pi$ increases, $\alpha$ needs to take larger values to satisfy the safety constraint, and this reduces the value of maximal improvement, $\zeta^+$.

Conversely, to attain a larger improvement $\zeta^+$, if we choose a smaller $\alpha$ for a problem where $C_{\mathrm{diff}}^\pi$ is large for all policies, then, we must give up on the safety guarantee, and $\zeta$ would be large. This means that the learned policy $\pi_\alpha$ might substantially degrade beyond the behavior policy.

## B.3 PROOF OF THEOREM 3.1

In order to prove this result, we we will utilize a basic algebraic inequality mentioned below in Lemma B.1.

**Lemma B.1** (Algebraic variation). *Given any $N$ positive real numbers, $x_1, x_2, \cdots, x_N$:*

$$\left( \sum_{i=1}^{N} \sqrt{x_i} \right)^2 \geq \sum_{i=1}^{N} x_i \geq \frac{1}{(N-1)} \sum_{i<j} (\sqrt{x_i} - \sqrt{x_j})^2. \tag{15}$$

*Proof.* For every $x_i$, define $y_i = \sqrt{x_i}$. Then, the difference between the two sides in the equation above is given by:

$$\sum_i y_i^2 - \frac{1}{(N-1)} \sum_{i<j} (y_i^2 + y_j^2 - 2y_i y_j) = \sum_i y_i^2 - \frac{N-1}{N-1} y_i^2 + \frac{1}{N-1} \sum_{i<j} 2y_i y_j \tag{16}$$

$$= \frac{1}{N-1} \sum_{i<j} 2y_i y_j \geq 0, \tag{17}$$

where the first step follows by rearranging $y_i$ and $y_j$, and the final step follows by noting that $y_i \geq 0$ for all $i$. The other inequality follows trivially by noting that $\sqrt{x_i}$ are positive, and applying the standard formula for sum of squares. $\square$

We will also require a Lemma that allows us to upper bound the performance difference $J(\pi) - J(\pi_\beta)$ in terms of the metric $D_{\text{CQL}}(\pi, \pi_\beta)$ that appears in the safe policy improvement guarantee in Equation 13.

**Lemma B.2** (Tight upper bound on policy improvement.). *Assume that the reward function $r(\mathbf{s}, \mathbf{a})$ of the MDP is bounded such that $\forall \mathbf{s}, \mathbf{a}, r(\mathbf{s}, \mathbf{a}) \in [-R_{\max}, R_{\max}]$. For any two policies $\pi$ and $\pi_\beta$, we have the following:*

$$J(\pi) - J(\pi_\beta) \lesssim \mathcal{O}\left( \frac{1}{(1-\gamma)^2} \right) \cdot \mathbb{E}_{\mathbf{s} \sim d^\pi} \left[ D(\pi(\cdot|\mathbf{s}), \pi_\beta(\cdot|\mathbf{s})) \right] \cdot R_{\max}. \tag{18}$$

*Proof.* The core of the proof of this lemma relies on the fact that for any given function $\nu(\mathbf{x})$ over some space $\mathbf{x}$, we can upper bound, $\Delta_\nu(p, q) := \mathbb{E}_{\mathbf{x} \sim p}[\nu(\mathbf{x})] - \mathbb{E}_{\mathbf{x} \sim q}[\nu(\mathbf{x})]$ in terms of $D(p, q)$. To show this, we expand this expression:

$$\Delta_\nu(p, q) := \sum_{\mathbf{x}} (p(\mathbf{x}) - q(\mathbf{x})) \cdot \nu(\mathbf{x}) \tag{19}$$

$$= \sum_{\mathbf{x}} q(\mathbf{x}) \cdot \left( \frac{p(\mathbf{x})}{q(\mathbf{x})} - 1 \right) \cdot \nu(\mathbf{x}) \tag{20}$$

$$= \sum_{\mathbf{x}: \frac{p(\mathbf{x})}{q(\mathbf{x})} \geq 1, \nu(\mathbf{x}) \geq 0} q(\mathbf{x}) \left( \frac{p(\mathbf{x})}{q(\mathbf{x})} - 1 \right) \nu(\mathbf{x}) + \sum_{\mathbf{x}: \frac{p(\mathbf{x})}{q(\mathbf{x})} < 1, \nu(\mathbf{x}) \geq 0} q(\mathbf{x}) \left( \frac{p(\mathbf{x})}{q(\mathbf{x})} - 1 \right) \nu(\mathbf{x})$$

$$+ \sum_{\mathbf{x}: \frac{p(\mathbf{x})}{q(\mathbf{x})} \geq 1, \nu(\mathbf{x}) \leq 0} q(\mathbf{x}) \left( \frac{p(\mathbf{x})}{q(\mathbf{x})} - 1 \right) \nu(\mathbf{x}) + \sum_{\mathbf{x}: \frac{p(\mathbf{x})}{q(\mathbf{x})} < 1, \nu(\mathbf{x}) \leq 0} q(\mathbf{x}) \left( \frac{p(\mathbf{x})}{q(\mathbf{x})} - 1 \right) \nu(\mathbf{x}).$$

Each of the four terms above can be bounded independently as follows: for the first two terms, we multiply by $\frac{p(\mathbf{x})}{q(\mathbf{x})}$, the third term is clearly negative, and the final term is upper bounded by

multiplying by $1 + \frac{p(\mathbf{x})}{q(\mathbf{x})}$:

$$\sum_{\mathbf{x}: \frac{p(\mathbf{x})}{q(\mathbf{x})} \geq 1, \nu(\mathbf{x}) \geq 0} q(\mathbf{x}) \left( \frac{p(\mathbf{x})}{q(\mathbf{x})} - 1 \right) \nu(\mathbf{x}) \leq \sum_{\mathbf{x}: \frac{p(\mathbf{x})}{q(\mathbf{x})} \geq 1, \nu(\mathbf{x}) \geq 0} q(\mathbf{x}) \frac{p(\mathbf{x})}{q(\mathbf{x})} \left( \frac{p(\mathbf{x})}{q(\mathbf{x})} - 1 \right) \nu(\mathbf{x}) \tag{21}$$

$$\sum_{\mathbf{x}: \frac{p(\mathbf{x})}{q(\mathbf{x})} < 1, \nu(\mathbf{x}) \geq 0} q(\mathbf{x}) \left( \frac{p(\mathbf{x})}{q(\mathbf{x})} - 1 \right) \nu(\mathbf{x}) \leq \sum_{\mathbf{x}: \frac{p(\mathbf{x})}{q(\mathbf{x})} \leq 1, \nu(\mathbf{x}) \geq 0} q(\mathbf{x}) \frac{p(\mathbf{x})}{q(\mathbf{x})} \left( \frac{p(\mathbf{x})}{q(\mathbf{x})} - 1 \right) \nu(\mathbf{x}) \tag{22}$$

$$\sum_{\mathbf{x}: \frac{p(\mathbf{x})}{q(\mathbf{x})} \geq 1, \nu(\mathbf{x}) \leq 0} q(\mathbf{x}) \left( \frac{p(\mathbf{x})}{q(\mathbf{x})} - 1 \right) \nu(\mathbf{x}) \leq 0 \tag{23}$$

$$\sum_{\mathbf{x}: \frac{p(\mathbf{x})}{q(\mathbf{x})} < 1, \nu(\mathbf{x}) \leq 0} q(\mathbf{x}) \left( \frac{p(\mathbf{x})}{q(\mathbf{x})} - 1 \right) \nu(\mathbf{x}) \leq \sum_{\mathbf{x}: \frac{p(\mathbf{x})}{q(\mathbf{x})} < 1, \nu(\mathbf{x}) \leq 0} q(\mathbf{x}) \left( \frac{p(\mathbf{x})}{q(\mathbf{x})} - 1 \right) \left( 1 + \frac{p(\mathbf{x})}{q(\mathbf{x})} \right) \nu(\mathbf{x}). \tag{24}$$

Finally, for each of these terms, we can now upper bound $\nu(\mathbf{x})$ by its maximum absolute value, $|\nu(\mathbf{x})| \leq \nu_0$, and combine the terms to get the following bound on $\Delta_\nu(p, q)$:

$$\Delta_\nu(p, q) \leq \nu_0 \sum_{\mathbf{x}: \nu(\mathbf{x}) > 0} p(\mathbf{x}) \left( \frac{p(\mathbf{x})}{q(\mathbf{x})} - 1 \right) + 0 + \nu_0 \sum_{\mathbf{x}: \frac{p(\mathbf{x})}{q(\mathbf{x})} < 1, \nu(\mathbf{x}) < 0} \left( \frac{p^2(\mathbf{x})}{q(\mathbf{x})} - q(\mathbf{x}) \right) \tag{25}$$

$$\leq \nu_0 \left[ \sum_{\mathbf{x}} \frac{p^2(\mathbf{x})}{q(\mathbf{x})} - 1 \right], \tag{26}$$

where Equation 26 follows from using the fact that for the case when $p(\mathbf{x})/q(\mathbf{x}) > 1$ but $\nu(\mathbf{x}) \leq 0$, $p(\mathbf{x}) \left( \frac{p(\mathbf{x})}{q(\mathbf{x})} - 1 \right) > 0$, and hence it upper bounds the RHS of Equation 23. For the last case, where $\mathbf{x}: \frac{p(\mathbf{x})}{q(\mathbf{x})} < 1, \nu(\mathbf{x}) < 0$, we note that $\sum_{\mathbf{x}: \frac{p(\mathbf{x})}{q(\mathbf{x})} < 1, \nu(\mathbf{x}) < 0} q(\mathbf{x}) \geq \sum_{\mathbf{x}: \frac{p(\mathbf{x})}{q(\mathbf{x})} < 1, \nu(\mathbf{x}) < 0} p(\mathbf{x})$, and hence the upper bound on this term in Equation 26 follows. To complete the argument note that $D_{\text{CQL}}$ exactly takes the form obtained in the final equation, and hence:

$$\Delta_\nu(p, q) \leq \nu_0 \cdot D(p, q).$$

We can now use this result to bound the return differences, by using standard results for bounding the performance difference between policies (Achiam et al., 2017; Schulman et al., 2015) in terms of $\frac{1}{(1-\gamma)^2} \times D(\pi, \pi_\beta)$. At the core of these results is a bound on $\mathbb{E}_{\mathbf{a} \sim \pi(\cdot|\mathbf{s})}[f(\mathbf{s}, \mathbf{a})] - \mathbb{E}_{\mathbf{a} \sim \pi_\beta(\cdot|\mathbf{s})}[f(\mathbf{s}, \mathbf{a})]$, and hence the result proven above directly applies. This proves the required result. $\quad\square$

## B.4 Additional Technical Lemmas

In this section, we will provide proofs of two technical lemmas, that allow us to conclude the proof of Theorem 3.1. For these lemmas, we will consider a generic optimization problem,

$$\max_{\mathbf{x}} \ f(\mathbf{x}) + \alpha \, g(\mathbf{x}), \tag{27}$$

where $g(\mathbf{x}) > 0$ for any $\mathbf{x}$, and $\alpha$ can only take non-negative values.

**Lemma B.3** (Value of $g(\mathbf{x})$ as a function of $\alpha$). *Let $\mathbf{x}_\alpha^*$ be the value of $\mathbf{x}$ that maximizes Equation 27 for a given fixed value of $\alpha$. Then the following statements hold:*

1. *For any $\alpha \geq \beta \geq 0$, $g(\mathbf{x}_\alpha^*) \geq g(\mathbf{x}_\beta^*)$.*

2. *For any $\alpha \geq \beta \geq 0$, $\alpha g(\mathbf{x}_\alpha^*) \geq \beta g(\mathbf{x}_\beta^*)$.*

*Proof.* For any given $\alpha$, $\mathbf{x}_\alpha^*$ satisfies the following inequality:

$$\forall \mathbf{x}', \ f(\mathbf{x}_\alpha^*) + \alpha g(\mathbf{x}_\alpha^*) \geq f(\mathbf{x}') + \alpha g(\mathbf{x}'). \tag{28}$$

Using the above relation, we can write down two inequalities relating $\mathbf{x}_\alpha^*$ and $\mathbf{x}_\beta^*$:

$$f(\mathbf{x}_\alpha^*) + \alpha g(\mathbf{x}_\alpha^*) \geq f(\mathbf{x}_\beta^*) + \alpha g(\mathbf{x}_\beta^*) \tag{29}$$

$$f(\mathbf{x}_\beta^*) + \beta g(\mathbf{x}_\beta^*) \geq f(\mathbf{x}_\alpha^*) + \beta g(\mathbf{x}_\alpha^*) \tag{30}$$

Now, adding the two inequalities above, and cancelling the terms $f(\mathbf{x}_\alpha^*) + f(\mathbf{x}_\beta^*)$ from both sides, we obtain:

$$(\alpha - \beta)\, g(\mathbf{x}_\alpha^*) \geq (\alpha - \beta)\, g(\mathbf{x}_\beta^*). \tag{31}$$

Since $\alpha - \beta$ is non-negative, it is either equal to 0, in which case $g(\mathbf{x}_\alpha^*) = g(\mathbf{x}_\beta^*)$ or it is positive, in which case, $g(\mathbf{x}_\alpha^*) \geq g(\mathbf{x}_\beta^*)$. Combining these two cases, we get the desired result in (1). For proving the second part (2), note that we can write the difference of the two sides as:

$$\alpha g(\mathbf{x}_\alpha^*) - \beta g(\mathbf{x}_\beta^*) = \alpha g(\mathbf{x}_\alpha^*) - \alpha g(\mathbf{x}_\beta^*) + \alpha g(\mathbf{x}_\beta^*) - \beta g(\mathbf{x}_\beta^*) \tag{32}$$

$$= \alpha \left[ g(\mathbf{x}_\alpha^*) - g(\mathbf{x}_\beta^*) \right] + (\alpha - \beta) g(\mathbf{x}_\beta^*) \tag{33}$$

$$\geq 0 + 0 = 0, \tag{34}$$

where the last inequality follows from the fact that $g(\mathbf{x}) \geq 0$ and $\alpha > \beta$. $\qquad\square$

We will now provide a proof for Theorem 3.1.

**Theorem B.2** (Theorem 3.1 restated). *W.h.p.* $\geq 1 - \delta$, *for any prescribed level of safety* $\zeta$, *the maximum possible policy improvement over choices of* $\alpha$, $J(\pi_\alpha) - J(\pi_\beta) \leq \zeta^+$, *where* $\zeta^+$ *is given by:*

$$\zeta^+ := \max_\alpha \quad h^*(\alpha) \cdot \frac{1}{(1-\gamma)^2} \quad s.t. \quad \frac{c_1}{(1-\gamma)^2}\sqrt{\frac{C_{diff}^{\pi_\alpha}}{|\mathcal{D}|}} - \frac{\alpha}{1-\gamma}\mathbb{E}_{\mathbf{s}\sim\widehat{d}^{\pi_\alpha}}\left[D(\pi_\alpha, \pi_\beta)(\mathbf{s})\right] \leq \zeta, \tag{35}$$

*where* $h^*$ *is a monotonically decreasing function of* $\alpha$, *and* $h(0) = \mathcal{O}(1)$.

***Proof of Theorem B.2.*** To prove this theorem, we will apply Lemma B.1 on $x_i = \frac{D(\pi_\alpha(\cdot|\mathbf{s}_i)\|\pi_\beta(\cdot|\mathbf{s}_i))}{\mu(\mathbf{s}_i)}$ and combine it with a the safe policy improvement guarantee for behavior regularization methods that admit updates of the form shown in Equation 1.

First, we note by applying Lemma B.1 in its expectation form that:

$$\left(\mathbb{E}_{\mathbf{s}\sim\widehat{d}^{\pi_\alpha}}\left[\sqrt{\frac{D(\pi_\alpha(\cdot|\mathbf{s})\|\pi_\beta(\cdot|\mathbf{s}))}{\mu(\mathbf{s})}}\right]\right)^2 \geq \mathbb{E}_{\mathbf{s}_1\sim\widehat{d}^{\pi_\alpha}, \mathbf{s}_2\sim\widehat{d}^{\pi_\alpha}}\left[\sqrt{\frac{D(\pi_\alpha(\cdot|\mathbf{s}_1)\|\pi_\beta(\cdot|\mathbf{s}_1))}{\mu(\mathbf{s}_1)}} - \sqrt{\frac{D(\pi_\alpha(\cdot|\mathbf{s}_2)\|\pi_\beta(\cdot|\mathbf{s}_2))}{\mu(\mathbf{s}_2)}}\right]^2,$$

where the term on the RHS of the above equation corresponds to $C_{diff}^\pi$.

Now we can plug this into the safe-policy improvement guarantee to obtain the resulting result as follows. Note that the first term in the bound in Equation 13 can be lower bounded using the differential concentrability as discussed above, and therefore, we get the following lower bound on $\zeta$:

$$\zeta \geq \mathcal{O}\left(\frac{1}{(1-\gamma)^2}\right)\sqrt{\frac{C_{diff}^{\pi_\alpha}}{|\mathcal{D}|}} + \alpha\mathbb{E}_{\mathbf{s}\sim\widehat{d}^{\pi_\alpha}}\left[D(\pi_\alpha, \pi_\beta)(\mathbf{s})\right], \tag{36}$$

which is exactly the same as the expression for the constraint in Theorem 3.1.

Next we provide an upper bound on the maximal improvement that can be possible, in terms of $D(\pi_\alpha, \pi_\beta)$. For this, we will utilize Lemma B.2, and we can directly upper bound $J(\pi_\alpha) - J(\pi_\beta)$ as follows:

$$J(\pi_\alpha) - J(\pi_\beta) \lesssim \frac{1}{(1-\gamma)^2}\mathbb{E}_{\mathbf{s}\sim\widehat{d}^{\pi_\alpha}}\left[D(\pi_\alpha, \pi_\beta)(\mathbf{s})\right] \cdot R_{\max}. \tag{37}$$

Finally, we express this upper bound in terms of $\alpha$.

Now, note that the RHS in Equation 37 depends on $\mathbb{E}_{\mathbf{s}\sim\widehat{d}^{\pi_\alpha}}\left[D(\pi_\alpha, \pi_\beta)(\mathbf{s})\right]$, which is directly the term that a generic distributional constraint algorithm minimizes (Equation 1). We wish to understand how this term evolves a function of $\alpha$.

Now we will invoke Lemma B.3 to understand the behavior of the term above when solving the optimization problem in Equation 1. To do so, consider any $\alpha$, $\alpha'$ and note that $f(\mathbf{x}) = \widehat{J}(\pi)$ and $g(\mathbf{x}) = -\mathbb{E}_{\mathbf{s} \sim \widehat{d}^\pi}[D(\pi, \pi_\beta)(\mathbf{s})]$. Now applying Lemma B.3, Part (1), we note that:

$$\mathbb{E}_{\mathbf{s} \sim \widehat{d}^{\pi_\alpha}(\mathbf{s})}[D(\pi_\alpha, \pi_\beta)(\mathbf{s})] \leq \mathbb{E}_{\mathbf{s} \sim \widehat{d}^{\pi_{\alpha'}}(\mathbf{s})}[D(\pi_{\alpha'}, \pi_\beta)(\mathbf{s})], \tag{38}$$

for $\alpha' \leq \alpha$. This means that we can upper bound this quantity by a function $h^*(\alpha)$ that is monotonically decreasing in $\alpha$.

Therefore, the maximal improvement is upper bounded by: $h^*(\alpha)\mathcal{O}\left(\frac{1}{(1-\gamma)^2}\right)$, which completes the proof of this theorem. $\qquad\square$

### B.5 PROOF OF LEMMA 4.1

**Lemma B.4** ((Lemma 4.1 restated) Per-state modification of Q-values.)**.** *Let $g$ represents* $g\left(\tau \cdot \pi(\cdot|\mathbf{s})\right)$. *The Q-function obtained after one TD-learning iteration using the objective in Eq. 10 is:*

$$Q_\theta(\mathbf{s}, \mathbf{a}) := \mathcal{B}^\pi \bar{Q}(\mathbf{s}, \mathbf{a}) - \alpha \frac{\pi(\mathbf{a}|\mathbf{s}) + \pi_\beta(\mathbf{a}|\mathbf{s})g - 2\pi_\beta(\mathbf{a}|\mathbf{s})}{2\pi_\beta(\mathbf{a}|\mathbf{s})} \tag{39}$$

*where $\mathcal{B}^\pi \bar{Q}(\mathbf{s}, \mathbf{a})$ is the Bellman backup operator applied to a delayed target Q-network.*

Recall from Section 2 that the objective of CQL consists of two terms

$$\min_\theta \ \alpha \underbrace{(\mathbb{E}_{\mathbf{s} \sim \mathcal{D}, \mathbf{a} \sim \pi}[Q_\theta(\mathbf{s}, \mathbf{a})] - \mathbb{E}_{\mathbf{s}, \mathbf{a} \sim \mathcal{D}}[Q_\theta(\mathbf{s}, \mathbf{a})])}_{\mathcal{R}(\theta)} + \frac{1}{2}\mathbb{E}_{\mathbf{s}, \mathbf{a}, \mathbf{s}' \sim \mathcal{D}}\left[\left(Q_\theta(\mathbf{s}, \mathbf{a}) - \mathcal{B}^\pi \bar{Q}(\mathbf{s}, \mathbf{a})\right)^2\right], \tag{40}$$

where $\mathcal{B}^\pi \bar{Q}(\mathbf{s}, \mathbf{a})$ is the Bellman backup operator applied to a delayed target Q-network. In tabular setting, the Q-function obtained after one iteration of TD-learning using the objective function in Eq. 40 is given by:

$$Q_\theta(\mathbf{s}, \mathbf{a}) := \mathcal{B}^\pi \bar{Q}(\mathbf{s}, \mathbf{a}) - \alpha \left[\frac{\pi(\mathbf{a}|\mathbf{s})}{\pi_\beta(\mathbf{a}|\mathbf{s})} - 1\right]. \tag{41}$$

In CQL, the result in Eq. 41 is obtained by setting the derivative of the objective in Eq. 40 with respect to the Q-values to 0, and solve for $Q_\theta(\mathbf{s}, \mathbf{a})$ (Kumar et al., 2020).

We now restate the new regularizer introduced by ReDS, and the new objective function for the Q-function.

$$\mathcal{R}(\theta; \rho) = \left(\frac{1}{2}\mathbb{E}_{\mathbf{s} \sim \mathcal{D}, \mathbf{a} \sim \pi}[Q_\theta(\mathbf{s}, \mathbf{a})] + \frac{1}{2}\mathbb{E}_{\mathbf{s} \sim \mathcal{D}, \mathbf{a} \sim \rho}[Q_\theta(\mathbf{s}, \mathbf{a})] - \mathbb{E}_{\mathbf{s}, \mathbf{a} \sim \mathcal{D}}[Q_\theta(\mathbf{s}, \mathbf{a})]\right) \tag{42}$$

$$\min_\theta \quad J_Q(\theta) = \mathcal{R}(\theta; \rho) + \frac{1}{2}\mathbb{E}_{\mathbf{s}, \mathbf{a}, \mathbf{s}' \sim \mathcal{D}}\left[\left(Q_\theta(\mathbf{s}, \mathbf{a}) - \mathcal{B}^\pi \bar{Q}(\mathbf{s}, \mathbf{a})\right)^2\right] \tag{43}$$

Notice that the main difference between the original CQL objective in Eq. 40 and the new objective in Eq. 43 is the distribution with which we push down Q values. The objective in Eq. 40 pushes Q values down under the learned policy $\pi$, whereas the objective in Eq. 43 pushes Q values down under a mixture of $\pi$ and $\rho$, i.e. $\frac{1}{2}\pi + \frac{1}{2}\rho$. Since $\rho$ is parameterized by a neural network whose input does not contain the Q values, its gradient with respect to the Q values is 0. Additionally, since the mixture $\frac{1}{2}\pi + \frac{1}{2}\rho$ plays the same role in Eq.42 that $\pi$ plays in the objective function in Eq. 40, we therefore can obtain the solution for the Q-values after updating the Q-function using the objective

function in Eq. 43 simply by replacing $\pi$ in Eq. 41 with the mixture. That is, in tabular setting, after updating the Q-function using the objective in Eq.43, the Q-values are:

$$
\begin{aligned}
Q_\theta(\mathbf{s}, \mathbf{a}) &= \mathcal{B}^\pi \bar{Q}(\mathbf{s}, \mathbf{a}) - \alpha \left[ \frac{\frac{1}{2}\pi(\mathbf{a}|\mathbf{s}) + \frac{1}{2}\rho(\mathbf{a}|\mathbf{s})}{\pi_\beta(\mathbf{a}|\mathbf{s})} - 1 \right]. \\
&= \mathcal{B}^\pi \bar{Q}(\mathbf{s}, \mathbf{a}) - \alpha \left[ \frac{\pi(\mathbf{a}|\mathbf{s}) + \rho(\mathbf{a}|\mathbf{s})}{2\pi_\beta(\mathbf{a}|\mathbf{s})} - 1 \right]. \\
&= \mathcal{B}^\pi \bar{Q}(\mathbf{s}, \mathbf{a}) - \alpha \left[ \frac{\pi(\mathbf{a}|\mathbf{s}) + \rho(\mathbf{a}|\mathbf{s}) - 2\pi_\beta(\mathbf{a}|\mathbf{s})}{2\pi_\beta(\mathbf{a}|\mathbf{s})} \right]. \\
&= \mathcal{B}^\pi \bar{Q}(\mathbf{s}, \mathbf{a}) - \alpha \left[ \frac{\pi(\mathbf{a}|\mathbf{s}) + \pi_\beta(\mathbf{a}|\mathbf{s})g(.) - 2\pi_\beta(\mathbf{a}|\mathbf{s})}{2\pi_\beta(\mathbf{a}|\mathbf{s})} \right].
\end{aligned}
$$

since $\rho$ subsumes $\pi_\beta \cdot g$, giving us the desired result.

### B.6   PROOF OF LEMMA 4.2

**Intuition: Comparison of CQL and CQL (ReDS) objectives** We will first intuitively compare the objectives in CQL and CQL (ReDS) to understand where the difference arises from. We write down the CQL (ReDS) objective below:

$$
\max_{\pi \in \Pi} \widehat{J}(\pi) - \frac{\alpha}{2(1-\gamma)} \mathbb{E}_{\mathbf{s} \sim \widehat{d}^\pi} \left[ D(\pi, \pi_\beta)(\mathbf{s}) + \cdot \mathbb{E}_{\mathbf{a} \sim \pi(\cdot|\mathbf{s})} \left[ g(\tau \cdot \pi(\mathbf{a}|\mathbf{s})) \, \mathbb{I}\{\pi_\beta(\mathbf{a}|\mathbf{s}) > 0\} \right] \right]. \tag{44}
$$

and now the CQL objective:

$$
\max_{\pi \in \Pi} \widehat{J}(\pi) - \frac{\alpha}{(1-\gamma)} \mathbb{E}_{\mathbf{s} \sim \widehat{d}^\pi} \left[ D(\pi, \pi_\beta)(\mathbf{s}) \right]. \tag{45}
$$

Observe that the only difference between CQL and ReDS stems from the fact that while the regularizer in CQL only optimizes the policy to stay close to the behavior policy, the regularizer in ReDS minimizes an additional term $\sum_\mathbf{a} \pi(\mathbf{a}|\mathbf{s})g(\tau \cdot \pi(\mathbf{a}|\mathbf{s}))$. This term attempts to make the behavior policy more "sharp", as $g$ is a monotonically decreasing function of its argument, by preventing $\pi$ to be less sharp in states where $\pi_\beta$ is broad. This enables $\pi$ to find the action in the dataset support that maximizes the learned return $\widehat{J}(\pi)$, even when $\pi_\beta$ is broad.

To prove Lemma 4.2, we will consider the following abstract update form for the policy evaluation version of CQL (ReDS), that obtains the next Q-function iterate $Q_{k+1}$:

$$
\min_Q \alpha \left( \mathbb{E}_{\mathbf{s} \sim \mathcal{D}, \mathbf{a} \sim \pi^{re}} [Q(\mathbf{s}, \mathbf{a})] - \mathbb{E}_{\mathbf{s} \sim \mathcal{D}, \mathbf{a} \sim \pi_\beta} [Q(\mathbf{s}, \mathbf{a})] \right) + \frac{1}{2} \mathbb{E}_{\mathbf{s}, \mathbf{a}, \mathbf{s}' \sim \mathcal{D}} \left[ (Q(\mathbf{s}, \mathbf{a}) - \mathcal{B}^\pi Q_k(\mathbf{s}, \mathbf{a}))^2 \right], \tag{46}
$$

**Lemma B.5** (CQL (ReDS) restated more completely.). *CQL (ReDS) solves the following optimization problem, when $\alpha$ is large enough:*

$$
\max_\pi \widehat{J}(\pi) - \frac{\alpha}{2(1-\gamma)} \mathbb{E}_{\mathbf{s} \sim \widehat{d}^\pi} \left[ D(\pi, \pi_\beta)(\mathbf{s}) + \mathbb{E}_{\mathbf{a} \sim \pi(\cdot|\mathbf{s})} \left[ g(\tau \cdot \pi(\mathbf{a}|\mathbf{s})) \, \mathbb{I}\{\pi_\beta(\mathbf{a}|\mathbf{s}) \geq 0\} \right] \right].
$$

*Proof.* For proving Lemma 4.2, we follow an argument similar to the proof of Theorem 3.1 from Kumar et al. (2020). By differentiating the above objective w.r.t. Q, we note that the ReDS + CQL objective above exhibits the following effective Bellman backup

$$
\forall \mathbf{s}, \mathbf{a} \in \mathcal{D}, Q_{k+1}(\mathbf{s}, \mathbf{a}) := (\mathcal{B}^\pi Q_k)(\mathbf{s}, \mathbf{a}) - \alpha \left( \frac{\pi^{re}(\mathbf{a}|\mathbf{s})}{\pi_\beta(\mathbf{a}|\mathbf{s})} - 1 \right). \tag{47}
$$

This backup is equivalent to running pessimistic RL with a reward bonus equal to $-\alpha \left( \frac{\pi^{re}(\mathbf{a}|\mathbf{s})}{\pi_\beta(\mathbf{a}|\mathbf{s})} - 1 \right)$, and therefore, the policy obtained by maximizing the resulting Q-function can

be expressed as:

$$\max_{\pi} \ \widehat{J}(\pi) - \alpha \frac{1}{1-\gamma} \mathbb{E}_{\mathbf{s} \sim \widehat{d}^{\pi}} \left[ \mathbb{E}_{\mathbf{a} \sim \pi(\cdot | \mathbf{s})} \left[ \left( \frac{\pi^{re}(\mathbf{a}|\mathbf{s})}{\pi_{\beta}(\mathbf{a}|\mathbf{s})} - 1 \right) \right] \right] \tag{48}$$

$$\equiv \max_{\pi} \ \widehat{J}(\pi) - \alpha \frac{1}{2(1-\gamma)} \mathbb{E}_{\mathbf{s} \sim \widehat{d}^{\pi}} [D(\pi, \pi_{\beta})(\mathbf{s})] - \frac{\alpha}{2(1-\gamma)} \mathbb{E}_{\mathbf{s}, \mathbf{a} \sim \widehat{d}^{\pi}} \left[ \mathbb{I}\{\pi_{\beta}(\mathbf{a}|\mathbf{s}) > 0\} g\left( \tau \cdot \pi(\mathbf{a}|\mathbf{s}) \right) \right]. \tag{49}$$

We retain the notion of support $\mathbb{I}\{\pi_{\beta}(\mathbf{a}|\mathbf{s}) > 0\}$ because $\pi_{\beta}(\mathbf{a}|\mathbf{s})$ in the denominator only cancels out for actions within the support of the behavior policy, as for all other actions, this term the term would be ill-defined as $\pi_{\beta}(a|s) = 0$ appears in the denominator. Hence this term in the above equation cannot sum up over actions $\mathbf{a}$ not in the behavior policy in the second term (these actions will be pushed down to have $-\infty$ Q-values). If we just cancelled the $\pi_{\beta}$ in the denominator, note that $g(\tau \cdot \pi(a|s))$ is not guaranteed to be 0 on actions $\mathbf{a}$, where $\pi_{\beta}(a|s) = 0$. Hence, we must retain the indicator function to explain this result properly. Instead, for the case of standard CQL, the divergence $D(\pi, \pi_{\beta})(s)$ term prevents $\pi$ from putting a non-zero density on actions where $\pi_{\beta}(a|s)$ in CQL, or else $D(\pi, \pi_{\beta})(\mathbf{s}) = \infty$, meaning that the Q-value for such a $\pi$ would be $-\infty$. $\qquad\square$

## C  IMPLEMENTATION DETAILS OF CQL (REDS)

In this section, we will provide implementation details about our algorithm, CQL (ReDS). The pseudo-code in Algorithm 1 illustrates the different update steps of our algorithms. In addition, we provided a detailed python-like algorithm description for ease of implementation. This can be found below in Section C.1.

Most of the components of Algorithm 1 are straightforward and follow the same convention, training update and, as we will discuss, hyperparameters as the CQL algorithm. This includes training the policy $\pi_{\phi}$, and for the most part training the critic $Q_{\theta}$. The main difference in the update for CQL (ReDS) is utilizing the mixture of $\pi$ and $\rho$ in the CQL regularizer. For obtaining $\rho_{\psi}$, we utilize a standard advantage-weighted training update, following the papers (Kostrikov et al., 2021a; Nair et al., 2020b; Peng et al., 2019). Following these prior works, we also clip the argument to the exponent between a minimum range and a maximum range to be numerically stable:

$$\mathbb{E}_{\mathbf{s}, \mathbf{a} \sim \mathcal{D}}[\log \rho_{\psi}(\mathbf{a}|\mathbf{s}) \cdot \exp\left[ \text{clip}\left( -A_{\theta}^{\pi}(\mathbf{s}, \mathbf{a})/\tau, \sigma_{\min}, \sigma_{\max} \right) \right]. \tag{50}$$

In our experiments, we chose $\sigma_{\min} = -10$ and $\sigma_{\max} = 5$ across all the tasks and domains we study. These details are standard in training advantage-weighted algorithms.

### C.1  DETAILED ALGORITHM DESCRIPTION FOR CQL (REDS)

Algorithm 1 provides the pseudo-code for CQL (ReDS). We provide the detailed description of how each update step in Algorithm 1 is implemented using Python syntax based on the PyTorch Framework in this section. We include 3 code listings below, illustrating the update steps for the parametric Q-functions, the policy and the learnt distribution $\rho$.

Listing 1: Training Q networks given a batch of data, corresponding to step 3 in Algorithm 1

```
q_data = critic(batch['observations'], batch['actions'])

next_dist = actor(batch['next_observations'])
next_pi_actions, next_log_pis = next_dist.sample()

target_qval = target_critic(batch['observations'],
                            next_pi_actions)
target_qval = batch['rewards'] + \
    self.gamma * (1 - batch['dones']) * target_qval

td_loss = mse_loss(q_data, target_qval)

# importance sampling term
num_samples = 4

# assume env is normalized between [-1, 1]
random_actions = uniform_sample((num_samples,
    batch['actions'].shape[-1]), min=-1, max=1)
random_pi = 0.5 ** batch['actions'].shape[-1]

dist = actor(batch['observations'])
pi_actions, log_pis = dist.sample(num_samples)

rho_dist = rho(batch['observations'])
rho_actions, log_probs_rho = rho_dist.sample(num_samples)

q_rand_is = critic(batch['observations'],
                   random_actions) - random_pi
q_pi_is = critic(batch['observations'],
                 pi_actions) - log_pis
q_rho_is = critic(batch['observations'],
                  rho_actions) - log_probs_rho

cat_q = concatenate(q_rand_is, q_pi_is, new_axis=True)
cat_q = logsumexp(cat_q, axis=-1)

cat_q_rho = logsumexp(q_rho_is, axis=-1)

# average between rho and pi
push_down_term_reds = 0.5 * (cat_q + cat_q_rho)

reds_loss = td_loss + \\
    ((push_down_term_reds - q_data).mean() * cql_alpha)

critic_optimizer.zero_grad()
reds_loss.backward()
critic_optimizer.step()
```

Listing 2: Training the policy (or the actor) given a batch of data (step 4 in Algorithm 1)

```
# Identical to CQL
# return distribution of actions
dist = actor(batch['observations'])

# sample actions with associated log probabilities
pi_actions, log_pis = dist.sample()

# calculate q value of actor actions
qpi = critic(batch['observations'], actions)
qpi = qpi.min(axis=0)

# same objective as CQL (kumar et al.)
actor_loss = (log_pis * self.alpha - qpi).mean()

# optimize loss
actor_optimizer.zero_grad()
actor_loss.backward()
actor_optimizer.step()
```

Listing 3: Training the $\rho_\psi$ distribution given a batch of data (step 5 in Algorithm 1)

```
# AWR style update to find rho

# sample policy actions for advantage calculation
dist = actor(batch['observations'])
pi_actions, log_pis = dist.sample()

# calculate advantage
qdata = critic(batch['observations'], batch['actions'])
value = critic(batch['observations'], actions)
advantage = (qdata - value.min(0)).mean()

# awr style clipping
clipped_advantage = clip(advantage/self.temperature, \
    min=-10, max=5)

# find log rho(a|s)
rho_dist = _rho(batch['observations'])
log_prob_rho = rho_dist.log_prob(batch['actions'])

# Advantage Weighted Log Probabilities is the loss for rho
rho_loss = -(exp(-clipped_advantage) * log_prob_rho)
rho_loss = rho_loss.mean()

rho_optimizer.zero_grad()
rho_loss.backward()
rho_optimizer.step()
```

# D   TASK AND DATASET DESCRIPTIONS

In this section, we will describe the various tasks we introduce in this paper. We also provide qualitative descriptions of these tasks here.

**Heteroskedastic antmaze navigation.** We introduce four new antmaze datasets which exhibit two different dataset distributions each for the medium and large mazes from D4RL (Fu et al., 2020). We reuse the layouts of the mazes directly from D4RL, but modify the data collection protocol. For the `noisy` datasets, given an observation from the environment, we first compute the action that would have been taken by the D4RL behavior policy, and then add Gaussian noise to the action. While this alone is not much harder, crucially, note that the variance of this added Gaussian noise differs depending on the location of the Ant in the 2D Maze. In addition there is a small bias added to the D4RL behavior policy, but this bias is dominated by noise. We present the noise standard deviations (indicated "Noise") and the bias added (indicated "Bias") for this dataset as a function of different location intervals in the maze in the left part of the Figure 8 below.

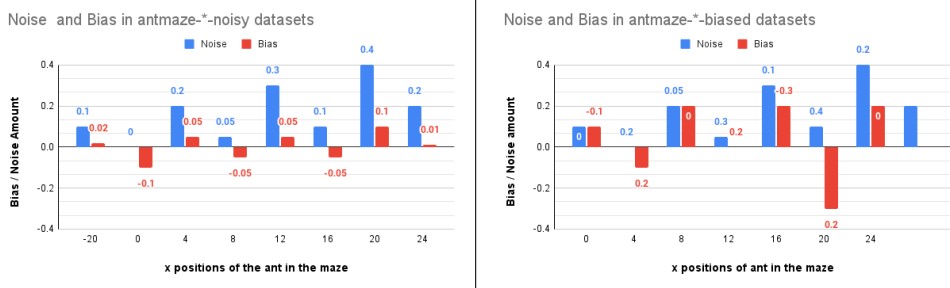

Figure 8: The distribution of noise and bias in the heteroskedastic antmaze datasets as a function of the x-position of the ant in the maze. While the noisy datasets primarily add noise, the biased datasets also add significant bias beyond the noise. The value of a given bar is the variance of the noise / bias added in the region between the x-position for that bar, and the next one.

For the `biased` datasets, in addition to adding location-dependent Gaussian noise to the action computed by D4RL behavior policies, we add a strong bias to the action (see Figure 8 (right)). Crucially note that the direction of this bias (i.e., the sign) changes based on the location of the Ant in the 2D maze, which mimics the scenario studied in our didactic navigation example in Section 3.1. In summary, because in some 2D regions of the maze, the values of the noise and bias added to the actions are larger, while in other 2D regions, they are smaller, the new offline datasets contain more heteroskedastic data distribution, where an optimal learned policy must deviate away from the data distribution much more in certain regions, whereas much lesser in other regions, which would correspond to an increase in the differential concentrability. This is demonstrated quantitatively in Table 4a. Thus, we expect that learning well on these tasks modulating the strength of the distributional constraints per state.

**Visual robotic pick and place.** We introduce a pick and place dataset, which exhibits a unique dataset distribution for a robotic pick and place manipulation task, building on the framework from Singh et al. (2020). As shown in Figure 10, the robotic setup is a 6-DOF WidowX robot in front of a green bowl with 2 objects: a target object (the ball in this case) and a distractor object (the can). The objective is to place the target object into the bin. The reward function is a sparse, binary indicator of success, where a +1 reward is given when the object is placed in the bin. This task must be done from $128 \times 128 \times 3$ raw visual observations, without access to either the robot state, or the state of the objects, which can change as the objects can roll on the surface.

**Visual robotic bin sorting.** We introduce a bin sorting tasks, which are also built on the framework from Singh et al. (2020). As shown in Figure 10, the robotic setup is a 6-DOF WidowX robot in front of two identical white bins with 2 objects to sort. The objective is to sort each object into its respective bins the target object in to the bin. The reward function is a sparse, binary indicator of success, where a +1 reward is given when both objects are placed in their correct bins. This task must be done from $128 \times 128 \times 3$ raw visual observations, without access to either the robot state, or the state of the objects, which can change as the objects can roll on the surface.

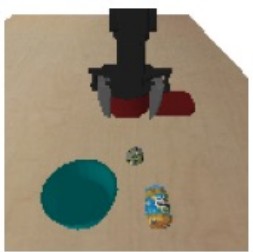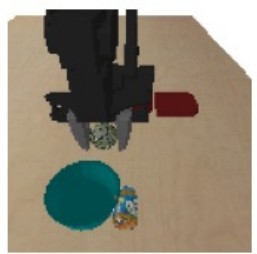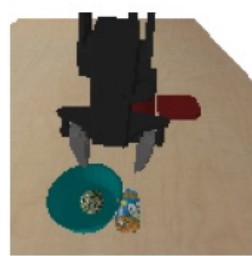

Figure 9: **Visualizing a sample trajectory for the visual pick-place robotic manipulation task.** Here is an example successful trajectory in the dataset collected using a scripted policy. Th robot reaches for the target object (the green ball), lifts it, and places it inside of the green container.

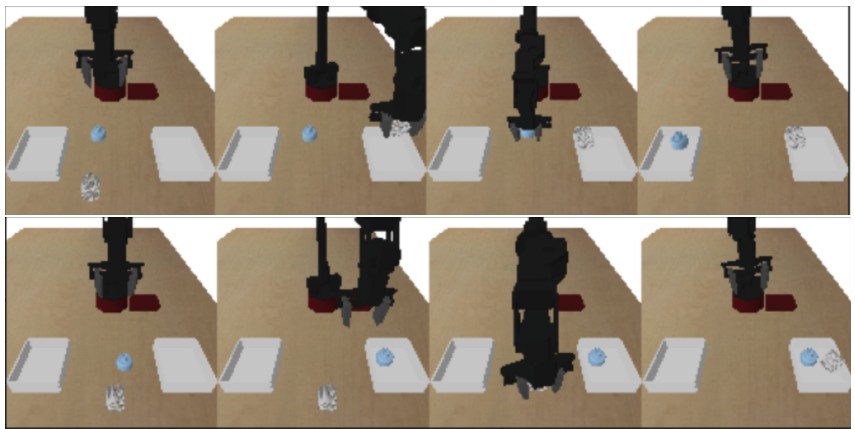

Figure 10: **Visualizing sample trajectories for the visual bin sorting robotic manipulation task.** Here are two sample trajectories for the binsorting domain. **Top:** A successful trajectory in the dataset. Here the robot places the white cylinder in the right bin and the blue ball in the left bin, succesfully sorting the objects into their respective bins. **Bottom:** A unsuccessful trajectory in the dataset. Here the robot places both objects in the same bin. This is unsuccessful as an object was placed in the incorrect bin thereby not sorting them in a correct manner.

To collect a heteroskedastic dataset, we run data collection using hardcoded scripted control policies, whose success rate and variance can be controlled. Each trajectory in the dataset was collected in the following manner. For the first phase, the robot reaches towards the object without any bias and grasps it with a reasonable success rate. Here, though, the noise and stochasticity in the scripted data collection and the inaccuracies in the scripted policy make it not succeed for every trial. During the second phase, where the robot places the object in the bin, there was a bias towards placing the object in a position in the workspace that does not correspond to the target location of the bin for which the robot can attain a reward, and which the robot will observe during evaluation. For our experiments, this bias was 85%. This forces the data distribution to be heteroskedastic: for the picking segment of this task, the behavior policy is centered around the desired optimal behavior i.e., grasping the object successfully, whereas for the placing segment, the behavior is biased towards carrying the object to the incorrect regions, requiring significant deviations from the behavior policy to succeed. An algorithm is now required to have a non-uniform amount of closeness to the behavior policy.

For the bin sorting domain, the easy domain had no bias and instead the scripted policy had additive gaussian white noise with a fixed variance which leads to low heteroskedacity.

**Atari game playing.** For the Atari tasks we consider in the paper, we devised a heteroskedastic data composition based on the DQN Replay dataset (Agarwal et al., 2020) which comprises of transitions found in the replay buffer of a run of an online DQN. Since this dataset consist of all the policies that the DQN agent produced over the course of training, and since Atari typically uses $\epsilon$-greedy exploration, where the value of $\epsilon$ decays over time, different trajectories of this dataset are generated

from different behavior policies, that all have different levels stochasticity. Naturally, since the value of $\epsilon$ decays over training and the performance of online DQN increases, the trajectories with higher return generally correlate with having lower stochasticity.

Given this information, we attempted to subsample a dataset that is heteroskedastic. For this purpose, we first divide the full replay buffer into $N$ equal chunks, where chunk 0 consists to experience observed earliest in training, while the chunk $N-1$ consists of experience seen near the final parts of training. Then, we subsample 20% of the trajectories from each of these chunks independently to obtain an intermediate dataset that comes from multiplies policies, observed at different times while training online DQN. Then, for any given trajectory $\tau$ of length $L$ in the replay chunk $i$, we only retain the transitions occurring between time steps $\lfloor \frac{(L-i)\times N}{L} \rfloor : \lfloor \frac{(L-i+1)\times N}{L} \rfloor$ in our final dataset and discard all the remaining transitions. This essentially means that the data closer to the initial states of the game comes from a good, less stochastic policy, whereas the data close to the final states of the game from a worse, highly stochastic policy. We develop two such datasets corresponding to $N=2$ and $N=5$ chunks. These chunks concatenated together construct the replay buffer of transitions that correspond to the 2 and 5 policy experiments seen in Section 5.

To see why this data is heteroskedastic, note that at different states of the game, we observe actions with different amounts of stochasticity and bias. This is because, as the game progresses, the effective behavior policy induced by the offline dataset exhibits a bias towards suboptimal actions (from the chunks that are earlier in DQN training) while also exhibiting substantial noise. The states that are closer to the initial states of the game, on the other hand, have an effective behavior policy that is primarily centered around a good action, with very little noise. In order to succeed, an offline RL algorithm must have different amount of conservatism at different states.

In our experiments, we considered 10 games including several standard games, and this is a subset of games studied in prior work (Kumar et al., 2021). The games we considered are: ASTERIX, BREAKOUT, Q*BERT, SEAQUEST, SPACEINVADERS, BEAMRIDER, MSPACMAN, WIZARDOF-WOR, JAMESBOND, PONG.

# E    EXPERIMENTAL DETAILS

For our experiments on the AntMaze domains, we built on the following open-source implementation of CQL: https://github.com/young-geng/JaxCQL, for our visual robotic experiments, we utilized our own port of the following implementation from Singh et al. (2020) in Jax: https://github.com/avisingh599/cog, and for our Atari experiments, we use the official implementation of CQL built on Dopamine (Castro et al., 2018): https://github.com/aviralkumar2907/CQL/tree/master/atari. For certain baselines (e.g EDAC, BEAR), we utilize the source implementation to stay consistent with the author's tested and tuned implementation. We additionally verified the results for D4RL benchmark for these tasks. We will summarize the hyperparameters in the next sections.

## E.1    HYPERPARAMETERS FOR CQL (REDS)

**Antmaze domains.** For the AntMaze domains, we utilized a temperature parameter $\tau = 0.3$ in our experiments (found by sweeping over $\tau \in [0.1, 0.3, 1.0, 5.0]$), for all the four dataset types in Table 4b. Every other hyperparameter was kept identical to CQL, which for the case of antmaze corresponds to applying the CQL regularizer $\mathcal{R}(\theta)$ with the dual version of CQL, where the threshold on the CQL regularizer is specified to be 1.0. Following CQL, we used 3-hidden layer critic and actor networks with layers of size 256, a critic learning rate of 3e-4 and an actor learning rate of 1e-4. We utilized the Bellman backup that computes the target value by performing a maximization over target values computed for $k = 10$ actions sampled from the policy at the next state.

**Atari domains.** For our Atari experiments, we tuned the value of $\alpha$ in CQL (Equation 2) between two values $[0.1, 0.2]$, and present the sensitivity results in Figure 4, and found that $\alpha = 0.1$ work better for CQL. We swept the value of $\tau \in [2.0, 5.0, 7.0]$ and report the sensitivity sweep in Figure 11.

**Visual Robotic Domains.** For the visual pick and place domains, we follow exactly the same hyperparameters as the CQL implementation from COG (Singh et al., 2020): a critic learning rate

of 3e-4, an actor learning rate of 1e-4, using $k = 4$ actions from the policy for computing the target values for computing the TD error, and using $k = 4$ actions to compute the log-sum-exp in CQL. For the value of $\tau$, we swept over $\tau \in [0.1, 1.0, 10.0, 100.0]$, and used a $\tau = 1.0$ for our experiments.

## F    ADDITIONAL ABLATION STUDIES

In this section, we present some results of an ablation study of the performance of CQL (ReDS) with respect to the temperature hyperparameter $\tau$ that appears in Equation 9. Before discussing the results, let us intuitively aim to understand the significance of this hyperparameter. When $\tau$ is extremely small we would expect $\rho_\psi$ to be a distribution centered at the worst possible action, within the support of the behavior policy. When $\tau$ is large, we would expect the learned $\rho_\psi$ to be close to the behavior policy, since the exponentiated advantage term would essentially behave as a constraint against a uniform distribution. Neither of these extremes are desirable, while the former does not behave much differently than a distributional constraint (except that the Q-value at the action with the smallest Q-value in the dataset support is not pushed up anymore), the latter also behaves like a distributional constraint, but with just half the effective multiplier $\alpha$ on the CQL regularizer. We would therefore expect an intermediate $\tau$ to perform the best.

To verify these insights, we study the sensitivity of the performance of CQL (ReDS) with respect to $\alpha$ on the Atari datasets. Our results shown in Figure 12 confirm that indeed an intermediate value of $\tau = 5.0$ out of the tested values, $\tau \in [2.0, 5.0, 7.0]$ works the best.

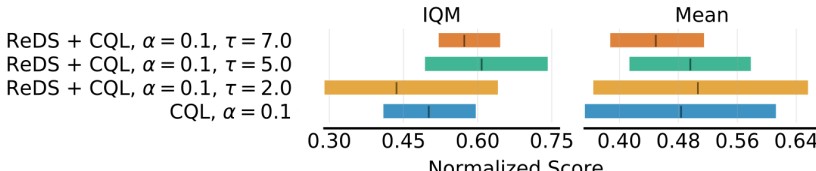

Figure 11: **Sensitivity of CQL (ReDS) to the temperature hyperparameter $\tau$ in Equation 9 evaluated on the Atari game experiments with 5 policies.** Observe that an intermediate value of temperature $\tau = 5.0$ works best

In addition, we study the sensitivity of the performance of CQL (ReDS) with respect to $\alpha$ on the Atari datasets. We report the performance for two different values of $\alpha \in \{0.1, 0.2\}$ from CQL (Equation 2) in Figure 12. Observe that CQL (ReDS) with a given $\alpha$ outperforms base CQL for the corresponding $\alpha$. Additionally note that the degradation in performance of CQL (ReDS) as $\alpha$ increases is lesser than base CQL.

## G    ADDITIONAL BASELINE COMPARISON FOR HETEROSKEDASTIC ANTMAZE NAVIGATION

In this section we will provide additional baseline comparison for REDS with two additional Offline RL methods: EDAC (An et al., 2021) and BEAR (Kumar et al., 2019).

### G.1    HYPERPARAMETERS FOR EDAC

As done in An et al. (2021), we tune the method over two hyperparameters. The first hyperparameter is the ensemble size $N$ which specifies the number of Q functions. The second parameter we consider is $\eta$, the weight of the ensemble gradient diversity term. Below in table 4, we show the values considered for each hyperparameter. There is significant overlap to these parameters with the ones used in the Mujoco Gym and Adroit Domains that the authors used. We utilized the publicly available code (https://github.com/snu-mllab/EDAC) released by the authors of EDAC and were able to replicate the results they reported for the D4RL MuJoCo Gym environments in An et al. (2021).

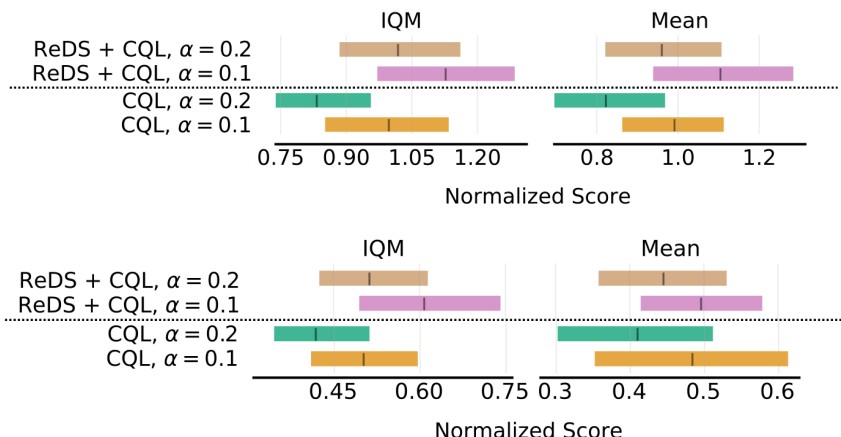

Figure 12: **Sensitivity of ReDS + CQL to the temperature hyperparameter** $\alpha$ **in Equation 2**
We report the performance of CQL (ReDS) vs CQL on the IQM normalized score and the mean
normalized score over ten Atari games, for the case of **two** (top) and **five** (bottom) policies. We
consider this performance for two different values of $\alpha \in \{0.1, 0.2\}$ in CQL (Equation 2). Observe
that CQL (ReDS) with a given $\alpha$ outperforms base CQL for the corresponding $\alpha$. Additionally note
that the degradation in performance of ReDS (CQL) as $\alpha$ increases is lesser than base CQL.

Table 4: EDAC Hyperparameters

| Hyperparameters | Values |
|---|---|
| $N$ | 10, 20, 50, 100 |
| $\eta$ | 0, 1, 5, 10, 50, 100, 1000 |

### G.2 HYPERPARAMETERS FOR BEAR

As done in Kumar et al. (2019), we tuned this method over two hyperparameters. The first
is the Kernel Type of the MMD between the behavior policy $\pi_\beta$ and the actor $\pi$, and found
that Laplacian performed better. The second parameter considered is $\sigma$, which is needed for
the Laplacian kernel as defined. Below in table 5, we show the values considered for each hy-
perparameter. There is significant overlap to these parameters with the ones used in the Mu-
joco Gym and Adroit Domains that the authors used. We utilized the publicly available code
(`https://github.com/rail-berkeley/d4rl_evaluations`) released by the authors
of BEAR and were able to replicate the results they reported for the D4RL MuJoCo Gym environ-
ments in Kumar et al. (2019).

Table 5: BEAR Hyperparameters

| Hyperparameters | Values |
|---|---|
| Kernel Type | Laplacian, Gaussian |
| $\sigma$ | 1, 10, 20, 50 |

