# OpenReview forum: "Offline Reinforcement Learning from Heteroskedastic Data Via Support Constraints"
_ICLR.cc/2023/Conference — Submitted to ICLR 2023_

### Official Review · Reviewer_HyhP · 2022-10-23

**Confidence:** 3
**Clarity, Quality, Novelty And Reproducibility:** The paper is clearly presented, and n…
**Correctness:** 4
**Technical Novelty And Significance:** 3
**Empirical Novelty And Significance:** 3
**Recommendation:** 6

**Strength And Weaknesses:**

The paper is well-written, theoretically motivated, and technically sound. I like Figure 1 which clearly illustrates the motivation of ReDS. However, I have two concerns.

First, how well does ReDS perform over non-heteroskedastic datasets?

Second, can ReDS be applied to offline datasets with a small number of samples? Do insufficient samples impose difficulty in estimating the distribution?


**Summary Of The Paper:**

The paper discovers that there may exist heteroskedasticity in realistic offline RL datasets, making current offline RL methods with distributional constraints suffer from performance degeneration. To address the problem, the paper proposes a novel method ReDS to convert distributional constraints into support-based constraints. Several new heteroskedastic datasets are constructed to demonstrate the superiority ReDS over current offline RL methods.

**Summary Of The Review:**

I recommend acceptance

---

> ### Author Response · Authors · 2022-11-13
> **Author Response**
>
> Thanks for your comments and for a positive assessment of our work!
>
> ___
>
> **ReDS with non-heteroskedastic datasets**
>
> We already evaluate ReDS on the standard D4RL datasets that are not heteroskedastic in Table 2, and find that the performance is similar to standard CQL (we also show that these datasets are not heteroskedastic by computing variability of behavior in Table 1(a)). In general, we would expect ReDS to recover the same performance as the underlying offline RL algorithm when the dataset is not heteroskedastic.
>
> Thus, when the dataset is heteroskedastic, we would expect that ReDS would perform better than CQL (as shown in the results in the paper), and when the dataset is non-heteroskedastic, both ReDS and CQL would perform the same. This means, that **in general, a support constraint algorithm like ReDS would be preferred**, especially if the dataset composition is not known.
>
> ___
>
> **Can ReDS be applied to offline datasets with a small number of samples?**
>
> Like any other offline RL method, we would expect that ReDS would also suffer from performance degradation when only provided with limited data. However, we do not expect that the overfitting errors in estimating the reweighted behavior policy, $\rho$ (Equation 9), would make the performance any worse than the standard CQL. Intuitively, this is because when the estimated distribution $\rho$ overfits to the training dataset, it would revert to becoming the indicator over the action $a$ observed at a state $s$ in the dataset, i.e., $\rho(a’|s) = \delta( a’ = a | (s, a) \in \mathcal{D})$. When this happens, the value of the regularizer $\mathcal{R}(\theta; \rho)$ (Equation 8) would revert back to the CQL regularizer, scaled by 0.5. To see this more formally, note that:
>
> $$\mathcal{R}(\theta; \pi_\beta) =  0.5  E_{ a \sim \pi}[Q(s, a)] + 0.5  E_{a \sim \rho}[Q(s, a)] - E_{a \sim \mathcal{D}}[Q(s, a)] = 0.5 E_{ a \sim \pi}[Q(s, a)] - 0.5  E_{a \sim \mathcal{D}}[Q(s, a)],$$
>
> which is the standard CQL regularizer itself. We can then simply scale up $\alpha$ in CQL to recover the same behavior from ReDS like standard CQL.
>
> ____
>
> **Please let us know if these answers address your concerns. If so, we would appreciate it if you would consider upgrading your score.**

---

> > ### Comment · Reviewer_HyhP · 2022-11-22
> > **Response**
> >
> > Thanks for your response that addresses part of my concerns. It may be a nontrivial limitation that ReDS outperforms CQL only over offline heteroskedastic datasets with sufficient samples, because plenty of samples are difficult to obtain in real-world RL problems.

---

### Official Review · Reviewer_d9Ww · 2022-10-25

**Confidence:** 4
**Correctness:** 2
**Technical Novelty And Significance:** 2
**Empirical Novelty And Significance:** 3
**Recommendation:** 6

**Clarity, Quality, Novelty And Reproducibility:**

The presentation quality seems to be very high, including a number of explanations with clear examples. However, technically, I do not agree with some of the ideas used to propose the algorithm. Theoretical results seem to be somewhat misleading. The experiment section is great. The algorithm is somewhat original, but it seems there are lots of ongoing researches that try to weight CQL regularization differently in each state.

**Strength And Weaknesses:**

Strengths
- High-quality descriptions with good examples
- Extensive evaluations over many different domains

Weaknesses
- Theoretical results are hard to follow and somewhat misleading:

1. Theorem 3.1 is especially hard to understand. While the policy improvement is shown with $\zeta^+$, the constraint is over $\zeta$, which makes it confusing because $\zeta$ and $\zeta^+$ do not seem to be related to each other in my opinion. And there is nothing about $\delta$ even though the theorem holds under prob. $1-\delta$. What is the variable that is constrained by the constraint? Is it $\alpha$ that is determined by the constraint? If it is the case, $J(\pi_\alpha)$ is determined by $\alpha$, which should not be used outside the maximization of eq (5). Overall, it is really hard to interpret the theorem.

2. Why do we need to use $g(\exp(A(s,a)))$ instead of $g(\pi(a|s)$? Can't we directly use $g(\pi(a|s))$ to weight the objective of eq (9)? It seems there isn't anything blocking us from doing such.

3. Lemma 4.2 seems to be quite misleading. Authors argue that we can put the notion of support with $\epsilon$ in the second term, but doesn't $\pi_\beta$ cancels out in the second term due to the definition of $\rho$? And the argument authors made seems to be also able to be applied on CQL case, since we can say that CQL regularization can be arbitrarily high when $\pi$ is high and $\pi_\beta$ is low. If it doesn't make CQL a support constraining algorithm, I think (ReDS) cannot be said to be a support constraining algorithm.

- Minor point: it seems gradient signs $\nabla_\psi, \nabla_\phi$ are omitted in Algorithm 1.

**Summary Of The Paper:**

This paper proposes an algorithm called CQL (ReDS), which uses reweighted data distribution to make the algorithm act like having a support constraint instead of a distribution constraint. For most of the complex offline RL problems, dataset distribution will be heteroskedastic, i.e. will contain different variances of actions in each state, and having a distribution constraint, in this case, can lead to a limited performance because we can always be overly pessimistic or optimistic for certain states. To prevent this, this paper show a simple way to achieve a support constraint instead with a small modification from original CQL algorithm, by using a mixed distribution $\pi^{re}$ instead of $\pi$ used to regularize in CQL.

$\pi^{re}$ is half $\pi$ and half $\rho$, where $\rho$ is some distribution of actions that is high when $\pi_\beta$ is high and $\pi$ is low, and $\rho$ will be low when $\pi_beta$ is low and $\pi$ is high. By having this additional $\rho$ term, policy $\pi$ of CQL(ReDS) can now be dramatically different from $\pi$. Experiment results show that by having this modified regularization the algorithm improves much.

**Summary Of The Review:**

Overall, I think the presentation quality is sufficient for being accepted. However, there are some points about the theory that is not clear to me, and I cannot recommend acceptance in its current form. I am ready to raise my score if those concerns are resolved.

----------
Reading the authors' response, I decided to raise the score. Thanks for the detailed response!

---

> ### Author Response · Authors · 2022-11-13
> **Author Response (Part 1 of 2): Summary; Clarifications on Theorem 3.1 and algorithm design**
>
> Thank you for your detailed feedback! To address your concerns regarding the theoretical statements, we have updated the paper to clarify the theoretical results and fixed issues with notation to ensure correctness. Our changes to the paper are shown in $\textcolor{magenta}{magenta}$. We explain our clarifications and corrections below as well.
>
> **We would appreciate it if you could have a look at our responses and let us know if the changes address your concerns. We are happy to clarify any other details further.**
>
> ___
>
> > ## Theorem 3.1 is hard to understand
>
> We have updated the statement and the discussion of Theorem 3.1 in the paper to clarify all the issues raised (please see Theorem 3.1 for the updates in magenta, and **Appendix B.2** for the formal re-statement and intuition).
>
> First, we clarify that the theorem bounds $\max_{\alpha} \left[J(\pi_\alpha) - J(\pi_\beta)\right]$ by $\zeta^{+}$, and so $\alpha$ does not appear outside in the maximization over $\alpha$ appearing in Equation 5.
>
> Second, the safety constraint restricts the value of $\alpha$ that can be used: the larger the value of $C^\pi_\text{diff}$ (for all policies), the larger the $\alpha$ required (as we discuss in intuition below **Theorem B.1**, and in **Lemma B.3**). As shown in Equations 37-38, the larger the $\alpha$, the smaller the $h^*(\alpha)$, which means that $\zeta^{+}$ is small for a given $\zeta$ if $C^\pi_\text{diff}$ is large.
>
> **Put in other words,** for any given value of differential concentrability, to attain a large $\zeta^+$, we must pick $\alpha$ as small as possible. If $C^\pi_\text{diff}$ is large, $\alpha$ can be made smaller only when $\zeta$ is large. **That is, a large $\zeta^{+}$ necessitates a larger $\zeta$, with heteroskedasticity.** Intuitively this means that to attain a larger improvement ($\zeta^{+}$), we must give up on the safety guarantee ($\zeta$).
>
> **$\delta$ does not appear:**  We apologize for the unclear notation:  $\delta$ was subsumed in the constant $c_1$. We have now added it explicitly as a logarithmic dependency in Theorem 3.1.
>
> ____
>
> > **Why do we need to use $g(\exp⁡(A(s,a)))$ instead of $g(\pi(a|s))$? Can't we directly use $g(\pi(a|s))$ to weight the objective of eq (9)? It seems there isn't anything blocking us from doing such**
>
> This is a great question! In theory, we can directly use $g(\pi(a|s))$ to weight the objective in Equation 9. The choice of utilizing the advantage function instead of the learned policy $\pi$ is largely a practical decision. Our intuition is that since $\rho$, the re-weighted behavior policy,  influences the choice of actions that the *current* Q-function is minimized under, using the current Q-function, instead of a **stale** estimate of the policy for deciding the up-weighting can perform better. Of course,  using $g(\exp(A(s, a)))$ instead of $g(\pi(a|s))$ is only a practical nuance since CQL updates the actor slower than the Q-function, while in theory, we could choose $g(\pi(a|s))$.
> ___

---

> > ### Author Response · Authors · 2022-11-13
> > **Author Response (Part 2 of 2): Clarifications on Lemma 4.2**
> >
> > > ## Clarifications regarding Lemma 4.2
> >
> > We have updated this Lemma in the paper, and its associated discussion in **Section 4.2** and **Appendix B.6** to address your concerns. To directly answer the questions asked, please note the following:
> >
> > **”Doesn’t $\pi_\beta$ cancel out?:”**
> >
> > It only cancels out for actions where $\pi_\beta(a|s) > 0$, or actions within the support of the behavior policy. As for all other actions, this term the term would be ill-defined since $\pi_\beta(a|s) = 0$ appears in the denominator. It would be incorrect to transform Equation 48 to Equation 49 without the indicator in Lemma 4.2,  as $g(\pi(a|s))$ is not guaranteed to be $0$ on actions $a$, where $\pi_\beta(a|s) = 0$. This argument is not applicable to the case of standard CQL because there is no such term like this, which only depends on $\pi$ (and not $\pi_\beta$). Instead, the divergence $D(\pi, \pi_\beta)(s)$ term prevents $\pi$ from taking a non-zero density on actions where $\pi_\beta(a|s)$ in CQL.  We clarify this in the paper, by writing down the equivalent optimization problems for CQL and ReDS side by side in Equations 44 and 45 (also copied below).
> >
> > ___
> >
> > **And the argument authors made seems to be also able to be applied on CQL case, since we can say that CQL regularization can be arbitrarily high when $\pi$ is high and $\pi_\beta$ is low. If it doesn't make CQL a support constraining algorithm, I think (ReDS) cannot be said to be a support constraining algorithm:**
> >
> > We think perhaps there might be a misunderstanding in what we mean by a “support constraint” algorithm and what our argument for ReDS is. A distributional constraint algorithm like CQL attempts to constrain $\pi$ to be close to $\pi_\beta$ in distribution. It also prevents $\pi$ from choosing any actions that lie outside of the support of $\pi_\beta$, but it overly constrains the learned policy $\pi$, preventing it from finding the **best in-support action.** A support constraint algorithm just constrains $\text{supp}(\pi) \subseteq \text{supp}(\pi_\beta)$, preventing $\pi$ from finding out-of-support actions, but does not constrain the distribution $\pi$ to $\pi_\beta$.
> >
> > Our explanation for why ReDS imposes an approximate constraint does not depend on the fact that the CQL regularizer is arbitrarily high or low. We first write down the objectives for CQL and ReDS:
> >
> > **ReDS**:
> >
> > $$\max_{\pi \in \Pi}~ \widehat{J}(\pi) - \frac{\alpha}{2 (1 - \gamma)} \mathbb{E}_{s \sim \widehat{d}^\pi} \left[ D(\pi, \pi_\beta)(s)  +
> >  \textcolor{magenta}{\mathbb{E}\_{a \sim \pi(\cdot|s)} \left[ g \left( \tau \cdot \pi(a|s) \right) \mathbb{I} \left[ \pi_\beta(a|s) > 0 \right]  \right]} \right]$$
> >
> > **CQL:**
> >
> > $$\max_{\pi \in \Pi}~ \widehat{J}(\pi) - \frac{\alpha}{(1 - \gamma)} \mathbb{E}_{s \sim \widehat{d}^\pi}\left[ D(\pi, \pi_\beta)(s) \right].$$
> >
> > Our argument follows the following line of logic:
> >
> > - The $D(\pi, \pi_\beta)(s)$ present in CQL encourages the policy to be close to the behavior policy $\pi_\beta$, i.e. $\pi \rightarrow \pi_\beta$.
> >
> > - The second-term in magenta, that only appears in ReDS essentially regularizes the policy to be “sharp” within the support over the behavior policy, it discourages the distribution of the learned policy $\pi$ from becoming constrained to the behavior policy.
> >
> > - **Behavior of CQL** On states where $\pi_\beta(\cdot|s)$ is quite broad (for simplicity, think of it as uniform over a subset of actions), the CQL regularizer would encourage $\pi$ to be broad as well, preventing it from concentrating on the best-in-support action (the regularizer is 0 if and only if, $\pi = \pi_\beta$). On states where $\pi_\beta(\cdot|s)$ is narrow, the CQL regularizer would encourage the learned policy to be narrow, mimicking the actions observed in the dataset.
> >
> > - **Behavior of ReDS** On states where $\pi_\beta(\cdot|s)$ is quite broad, the CQL regularizer (first term) would encourage $\pi$ to be broad, but the second term would encourage $\pi$ to be sharp, correcting some of the effects of the distributional constraint imposed by the CQL regularizer. In states where $\pi_\beta$ is narrow, both the terms would constrain $\pi$ to be closer to $\pi_\beta$. Due to the indicator in the second term, the policy $\pi$ is not encouraged to pick out-of-support actions in this case.
> >
> > **Thus, in summary,** our argument is that while the distributional constraint imposed by the CQL regularizer always encourages $\pi$ to match $\pi_\beta$, the constraint imposed by ReDS counters this effect of matching the behavior policy by utilizing the re-weighted distribution, enabling the policy $\pi$ to optimize the return $\widehat{J}(\pi)$ better. Thus, ReDS acts as an approximate support constraint.

---

> ### Author Response · Authors · 2022-11-16
> **Request for discussion**
>
> Dear Reviewer d9Ww,
>
> We were wondering if you have gotten the chance to go over our responses and the paper updates (in $\textcolor{magenta}{magenta}$). We would be eager to know if your concerns are addressed. We are happy to engage further and address any other concerns.
>
> Thanks a lot!!

---

### Official Review · Reviewer_N7S6 · 2022-10-25

**Confidence:** 4
**Clarity, Quality, Novelty And Reproducibility:** See above
**Correctness:** 3
**Technical Novelty And Significance:** 3
**Empirical Novelty And Significance:** 2
**Recommendation:** 5

**Strength And Weaknesses:**

## Pros
- Achieving support-based constraints is well-motivated and elegant. In fact, there exist support-based constraint methods, e.g., BEAR. But MMD constraint can not be accomplished in practice. Thus it performs poorly on the medium-expert dataset (with a wider action distribution from the medium-level policy). This paper implicitly achieves support-based constraints without estimating a behavior policy, which is elegant.

- The evaluation is conducted on four different types of datasets, including (1) D4RL Mujoco locomotion (*-medium-expert and *-medium-replay) datasets, (2) noisy/biased D4RL maze datasets, (3)the manipulation tasks, and (4) Atari games. The baseline algorithms, in my view, are also comprehensive.

## Concerns
- (Setting) The author says that the dataset for offline RL algorithms must illustrate the consequences of a diverse range of behaviors. As a result of this coverage, e.g., combining many realistic sources of data, algorithms should be able to learn from heteroskedastic datasets. But (1) I don't understand the logic of the argument. If such coverage benefits offline RL algorithms, why do we need to modify the existing offline RL algorithms for the dataset with good coverage? (2) Another concern is that recent work [1] shows that vanilla off-policy RL agents can outperform carefully designed ofﬂine RL algorithms with sufﬁciently diverse exploratory data. Could you please compare the heteroskedastic data and the diverse exploratory data? (3) Also, I am concerned about the motivation of the setting. The author clarifies the characteristics of heteroskedastic datasets, i.e., some dataset states correspond to narrow actions while others may be wider. But I can't easily see how this dataset relates to real-world applications.

- (Algorithm) The authors build their proposed algorithm on top of CQL, with improvements focused on implicit support based constraint. The modification on the CQL regular term is described in Eq. (8), which involves sampling over the re-weighted policy, \pi^{re}. I understand the idea that applying a distributional constraint on this re-weighting imposes a support based constraint. But I am not sure why the reweighted policy is deigned as a mixture of the learned policy \pi and the \rho, as described in Eq. 7. Could you please explain why the coefficient is designed as 0.5 and 0.5, or why not only use a \rho? By the way, the sampling strategy of Eq. 9 and the third line of the algorithm seem to be inconsistent.

- (Evaluation) The authors completed tests on three different datasets. Evaluations on the D4RL medium-expert and medium-replay datasets are really motivated, as these types of behavior policies can be common. But I think the dataset used in the second experiment, i.e., the noisy and biased D4RL antmaze dataset, is a bit contrived.

- (Minor)It could be better if the authors add a discussion about the scope of the method. The problem considered in this paper can be well summarized in Figure 3: when the actions corresponding to the dataset states contain both expert and non-expert decisions, the support-based constraint method should be used. I think this is quite true, as actions from experts and non-experts can be different. However, when only non-expert dataset actions are available for some dataset states, how does the support-based constraint approach work? This can be true, as non-expert actions often result in a transfer to a non-expert state that differs from the state visited by expert behavior policy. Finally, I want to say that I don't think this discussion detracts from the contribution of this paper. It is entirely up to the authors to decide whether or not to give such a discussion.

## Some typos.
- Page3, the first line, \mu should be replaced with \pi.
- Page6, under equation 7, Eq.3 should be Eq.2.
- Page6, under equation 11, Eq.30 should be Eq.11.
- Page7, ‘constrain the entropy of … to be uniform’.


[1] Don't Change the Algorithm, Change the Data: Exploratory Data for Offline Reinforcement Learning. https://arxiv.org/abs/2201.13425

**Summary Of The Paper:**

This paper considers a novel setting, i.e., learning policy from a heteroskedastic dataset with the demonstrated behaviors varies non-uniformly across states. For example, for a given maze environment and some non-optimal behavior policies, the states in the narrow hallways may exhibit a narrow action distribution, while more uniform actions for a wide room. In my view, this is true. In many cases, we cannot obtain the optimal policy with narrow action distribution for every state (if we can, we don’t need offline RL.). And suboptimal policies prefer to show different actions for a given state.

The authors propose empirical and theoretical evidence (Theorem 3.1) showing that explicit distributional constraints between the policy action and the (non-optimal) dataset actions may undermine the performance of the learned policy .

The authors try to achieve a support-based policy constraint to improve over the non-optimal behavior policies that generated the heteroskedastic datasets. That is, if the policy decision is within the support of the behavior policy, then the constraint should be zero. This constraint is weak than the distributional constraint. It thus holds the promise to derive a better policy on the states with more uniform action (non-expert) distributions.

To practically achieve the support constraint, the authors reweigh the behavioral policy into \pi_{re} and implicitly minimize D(\pi, \pi_{re}, where
$\pi_{re}(\cdot |s) = \frac{1}{2}\pi(\cdot \pi) + \frac{1}{2}[\pi_{beta}(\cdot |s) \cdot g(\exp(A(s, \cdot)))]$.

This work proposes an offline RL algorithm with support-based policy constraints. Although the evaluation is only based on the heteroskedastic dataset, I believe this pradgigm enjoys priority over the distributional constraint methods, such as the recent strong baseline algorithm, TD3+BC.


**Summary Of The Review:**

See above

---

> ### Author Response · Authors · 2022-11-13
> **Author Response (Part 1 of 3): Comparison to exploratory / diverse data**
>
> Thank you for your detailed feedback.  To briefly summarize our response, we clarify that heteroskedastic datasets are _distinct_ from exploratory datasets, and unlike exploratory data, they actually present a challenge for current offline RL algorithms (**Appendix A.2**). We also discuss how such heteroskedastic datasets appear in a variety of realistic scenarios – in robotic manipulation, in autonomous driving. We also address other concerns pertaining to the design of the algorithm (Section 4.1) and our evaluations (Section 5.2) one by one below. Our changes to the paper are shown in $\textcolor{magenta}{magenta}$.
>
> **We would really appreciate it if you could check our responses and let us know if your concerns are resolved. We are happy to clarify further if you have any other concerns and would really appreciate it if you engage in a discussion with us!**
>
> ___
>
> > ## Setting
>
> 1. **I don't understand the logic of the argument. If such coverage benefits offline RL algorithms, why do we need to modify the existing offline RL algorithms for the dataset with good coverage?**
>
> This is a great question! Heteroskedasticity is different from offline RL and we do not expect heteroskedasticity to help offline RL methods. Instead, we argue that it is a more challenging dataset composition that necessitates newer algorithms. We explain why below:
>
> In the context of offline RL, coverage typically refers to being able to cover **all** possible state-action pairs, as uniformly as possible. On the other hand, heteroskedasticity is used to refer to **disproportionate** coverage: when the action distribution in the offline dataset is more uniform at certain states, but very narrow at others. Thus, a heteroskedastic dataset does not have high coverage in the sense of observing all possible state-action pairs.
>
> Mathematically, while the notion of coverage is typically quantified by the concentrability coefficient which denotes the **worst case** density ratio between the distribution of the training dataset and the distribution of the learned policy, we quantify the notion of heteroskedasticity by the **variation** in density ratios across the state space. Even if the dataset has high coverage (i.e., it has a low concentrability coefficient), it can be heteroskedastic, since the variation in the density ratios across state spaces can lead to poorer performance.
>
> We have now updated the paper to add a discussion on coverage vs heteroskedasticity in Appendix A.2.
>
> ___
>
> 2. **Could you please compare the heteroskedastic data and the diverse exploratory data?**
>
> Most of the schemes used to generate exploratory data in Yarats et al. 2022 attempt to cover the state space (or state-action space) as uniformly as possible. For instance, the methods classified as “Data” by Yarats et al. 2021 (APT, ProtoRL) attempt to maximize state coverage, the methods classified as “Knowledge” (ICM, Disagreement, RND) attempt to find surprising state-action pairs and visit them, and the methods classified as “Competence” (SMM, DIAYN, APS) that maximize the mutual information between the state distribution attained by the learned policy and the learned skills also implicitly optimize for state coverage.
>
> In contrast, as discussed above, heteroskedasticity is not a condition of state coverage, it is only a condition of the variation of action coverage across different states.  A heteroskedastic dataset need not cover all states or all possible state-action pairs. It only exhibits disproportionate coverage of the action space in different states and is hence distinct from exploratory data. We have added this discussion in Appendix A.2.

---

> > ### Author Response · Authors · 2022-11-13
> > **Author Response (Part 2 of 3): Further clarifications on motivation; algorithm and evaluation**
> >
> > 3. **I am concerned about the motivation of the setting. The author clarifies the characteristics of heteroskedastic datasets, i.e., some dataset states correspond to narrow actions while others may be wider. But I can't easily see how this dataset relates to real-world applications. But I can't easily see how this dataset relates to real-world applications**
> >
> > To answer this question, we provide some real-world examples of datasets where the action distributions are very broad at some states while being very narrow at other states:
> >
> > (1) **In robotic manipulation tasks**, when considering datasets collected by human tele-operators, the action distribution on states encountered when approaching the object would be very diverse as different human tele-operators would try to navigate towards the object differently, but the action distribution at states encountered after an object is picked up would be very narrow as an arbitrary action would cause the object to fall off.
> >
> > (2) **In autonomous driving**, the behaviors of different human drivers while driving within a city vary a lot as everyone is navigating to different geographical locations and hence driving data from multiple drivers would induce a behavior policy that admits a broad action distribution at locations inside a city. On the other hand, the action distributions on a highway outside the city would be very narrow as every driver is attempting to go in one direction or the opposite.
> >
> > At an abstract level, the underlying theme in each of these examples is the existence of near-expert data at some states, where broad action distributions at other states.
> >
> > ___
> >
> > > ## Algorithm
> >
> > **But I am not sure why the reweighted policy is deigned as a mixture of the learned policy $\pi$ and $\rho$, as described in Eq. 7. Could you please explain why the coefficient is designed as 0.5 and 0.5, or why not only use a $\rho$?**
> >
> > Thanks for the question! Just utilizing $\rho$ would be insufficient, and would not be able to prevent the over-estimation of Q-values at out-of-support actions. This is because $\rho$ is only a re-weighting of the behavior policy (see Equation 6: $\rho(a|s) = \pi_\beta(a|s) g(a|s)$) and hence shares the same support as the behavior policy as well. If we only utilize $\rho$ in the regularizer $\mathcal{R}(\theta, \rho)$, then our algorithm will not exhibit any pessimism on the out-of-support actions, and will fail to learn a Q-function with lower values outside of the support of the behavior policy. We have now clarified on the role of using both $\pi$ and $\rho$ in **Section 4.1** in $\textcolor{magenta}{magenta}$.
> >
> > In the tabular setting, any mixture of $\pi$ and $\rho$ with a non-zero weight on $\pi$ will ensure that Q-values at out-of-support actions are controlled. With function approximation, however, we would need to put a significantly larger weight on $\pi$. In our experiments, we simply chose to use a weight of 0.5, but in principle, a better weighting can be derived by accounting for the generalization of the Q-function approximator.
> >
> > ___
> >
> > > ## Evaluation
> >
> > **Evaluations on the D4RL medium-expert and medium-replay datasets are really motivated, as these types of behavior policies can be common. But I think the dataset used in the second experiment, i.e., the noisy and biased D4RL antmaze dataset, is a bit contrived.**
> >
> > We have now updated Section 5.2 to clarify that the noisy and biased D4RL antmaze datasets are meant to be a stress test for our method.  That said, we present results of other tasks with realistic data compositions (1) visual pick and place, and (2) visual bin sort. The goal in these tasks is to manipulate objects. For instance, in the visual bin sort task, we must sort the objects in front of the robot into the correct bins. The offline dataset presents behaviors that vary in their diversity for different objects:  while the data shows certain objects being placed in either of the two bins in the data (diverse), others are placed only in one particular bin (narrow).
> >
> > A real-world motivating example of this would be sorting trash into a recycling bin and trash bin. Recyclable objects could potentially be sorted into either bin (diverse) but it would be better to place them in the recycling bin, whereas the non-recyclable objects must only be placed in the trash bin (narrow). If we potentially collect data of humans throwing away their lunch containers after they finish eating, we observe diverse behavior across all humans for the recyclable objects, for other objects we see narrow sets of actions of throwing them into the trash bin.

---

> > > ### Author Response · Authors · 2022-11-13
> > > **Author Response (Part 3 of 3): Minor concerns**
> > >
> > > > ## Minor
> > >
> > > **However, when only non-expert dataset actions are available for some dataset states, how does the support-based constraint approach work?**
> > >
> > > Offline RL methods attempt to find the best possible policy given the dataset, which need not be the optimal policy necessarily. Even when only non-expert actions are available, the support-based approach attempts to find the best action within support of the dataset at each state. As evidence, note that the noisy and biased antmaze datasets do not contain any expert trajectory in the dataset, yet our approach can find a good policy that succeeds in reaching the goal 73% of the time.

---

> ### Author Response · Authors · 2022-11-16
> **Request for discussion**
>
> Dear Reviewer N7S6,
>
> We were wondering if you have gotten the chance to go over our responses and the paper updates (in $\textcolor{magenta}{magenta}$). We would be eager to know if your concerns are addressed. We are happy to engage further and address any other concerns.
>
> Thanks a lot!!

---

> > ### Author Response · Authors · 2022-11-18
> > **Follow up**
> >
> > Dear Reviewer N7S6,
> >
> > Only a few hours are left in the discussion. We would really appreciate it if you could tell us if your concerns are resolved. Reviewer d9Ww found that their concerns were addressed, and we have attempted to address all of your concerns.
> >
> > We would be more than happy to resolve any remaining questions in the time we have, and would like to engage in a discussion.
> >
> > Thanks!

---

> > ### Comment · Reviewer_N7S6 · 2022-11-20
> > **Minor concerns**
> >
> > I still have two minor concerns about this work. The first is why the reweighted policy \pi_{re} is designed as a mixture of the learned policy $\pi$ and the reweighted behavior policy $\pi_{\beta}(a|s)g(a|s)})$. The authors have added some intuition to it. In my view, the design of \pi_{re} is the most important for this work, but it seems to lack formal analysis.
> >
> >
> > Another concern is the limitation of this work. An ideal support-based constraint is beneficial for policy improvement. But it does not help in the following setting. Non-expert actions often result in non-expert next-state that can only be visited by non-expert behavior policy. And, of course, only non-expert actions exist for these states. In this situation, I can not see how support-based constraint methods handle this.
> >
> > In summary, I am not sure this work's contribution reaches this conference's threshold, and the algorithm design analysis is somewhat lacking.

---

> > > ### Author Response · Authors · 2022-11-20
> > > **Author Response: Clarifications on minor concerns**
> > >
> > > Thank you for your response.
> > >
> > > Please let us know if our answers below resolve your concerns. If you think these are not sufficient, we would appreciate your help and engagement to enable us to address these concerns.
> > >
> > > > _The first is why the reweighted policy $\pi^\text{re}$ is designed as a mixture of the learned policy  and the reweighted behavior policy $\pi\_{\beta}(a|s)g(a|s)$. The authors have added some intuition to it. In my view, the design of \pi_{re} is the most important for this work, but it seems to lack formal analysis._
> > >
> > > **We believe all of our analysis in Section 4.2 (Lemmas 4.1 and 4.2) already provides a formal analysis of the performance guarantee for this choice of $\pi^\text{re}$**, and compares it to standard CQL, a distributional constraint algorithm. For instance, utilizing a mixture of $\pi$ and $\pi_\beta(a|s) g(a|s)$ enables us to obtain the two terms in Equation 12 in Lemma 4.2. A mixture of these two terms enables us to impose an approximate support constraint, whose benefits are discussed right below Lemma 4.2.
> > >
> > > We would also clarify the importance of having a mixture here. Formally, if we don’t utilize $\pi$ in $\pi^\text{re}$, then, we would not obtain the $D(\pi, \pi_\beta)(s)$ term in Equation 12. This would mean that the algorithm would not be conservative and would fail in cases when the behavior policy, $\pi_\beta$, is narrow. If we don’t utilize $\pi_\beta(a|s) g(a|s)$ in Equation 12, then the algorithm reverts to standard CQL, just with a different choice of $\alpha$ (note the $\frac{1}{2}$ multiplier on $\frac{\alpha}{1 - \gamma}$ in the Equation 12). **We will add this discussion in the next revision of the paper.**
> > >
> > > ___
> > >
> > > > _Another concern is the limitation of this work. An ideal support-based constraint is beneficial for policy improvement. But it does not help in the following setting. Non-expert actions often result in non-expert next-state that can only be visited by non-expert behavior policy. And, of course, only non-expert actions exist for these states. In this situation, I can not see how support-based constraint methods handle this._
> > >
> > > We might be misunderstanding your concern, so we would appreciate any clarification if the following answer does not seem sufficient.
> > >
> > > By this logic, if the offline dataset is generated using non-expert behavior policies, then no offline RL method should ever improve over the observed dataset. However, it is well known that offline RL methods do improve over the behavior policy even if it only consists of non-expert actions (for instance, empirically in D4RL tasks, methods do improve over the average dataset return, especially in the antmaze domains; offline RL methods such as CQL or Decision transformers improve over highly sub-optimal datasets in Atari). Intuitively, this happens because offline RL methods can combine or “stitch” together good parts of sub-optimal trajectories to find a much better policy. **In this paper, we show that support constraints are preferred for performing this improvement compared to distributional constraints, both theoretically and empirically.**
> > >
> > > We agree that there can be settings where our dataset consists of only non-expert actions and visits states that all attain poor rewards. For instance, if our dataset shows only failed trajectories in a robotic manipulation task. In such cases, no offline RL algorithm, support-constrained or not, will be able to find a better policy that improves over the behavior policy. This limitation is not specific to support constraints or our method, but to offline RL in general. In fact, more generally, no learning-based approach can find a good policy if we only ever had such failed data. **We believe that such a generic limitation of offline RL should not be used to evaluate our work.**
> > >
> > > Finally, we would like to clarify that we do not claim anywhere that support constraints will be able to learn the optimal policy given any kind of dataset.
> > >
> > > **Does this address your concern?**
> > >
> > > ___
> > >
> > > ***Please let us know if our response addresses your concerns. We hope that if your concerns are resolved, you would consider upgrading your score, and if not, you would engage with us further. We would be happy to resolve your remaining concerns.***

---

> > > > ### Author Response · Authors · 2022-11-22
> > > > **Does our response above answer your minor concerns?**
> > > >
> > > > Dear Reviewer N7S6,
> > > >
> > > > Thanks again for engaging with us. We responded to your minor concerns two days ago and are eagerly waitting to hear if your concerns are resolved. We would love to hear your thoughts on our responses above.
> > > >
> > > > Thanks a lot for discussing with us!

---

### Comment · Area_Chair_zroh · 2022-11-18
**Authors request feedback**

Dear reviewers,

The discussion period will end soon, and the authors have not received any feedback. If you have already read the responses from the reviewer and you do not plan to change your mind. You can ignore this message. However, if you have not read the response yet. Would please look at the response and give some feedback to the authors?

Best,
AC

---

### Decision · Program_Chairs · 2023-01-20

**Decision:**

Reject

**Justification For Why Not Higher Score:**

See weakness.

**Justification For Why Not Lower Score:**

N/A

**Metareview: Summary, Strengths And Weaknesses:**

This paper considers the problem of RL training on heteroskedastic off-line dataset. Instead of the classical distribution proximity regularizer w.r.t. the behavioral policy, the paper considers the support constraints that are more flexible to the considered setting. The motivation of this paper is clear and makes sense.

The weakness of the paper is mainly on the theoretical side. In this paper, the main theoretical contribution seems to be Theorem 3.1, which proves a limited policy improvement via distributional constraints. However, the upper bound on the improvement is the solution to some optimization problem but we do not have a clear characterization on how limited this improvement can be.

Moreover, the reviewer also think it may be a nontrivial limitation that ReDS outperforms CQL only over offline heteroskedastic datasets with sufficient samples, while plenty of samples are difficult to obtain in real-world RL problems.

**Summary Of Ac-Reviewer Meeting:**

The reviewers have not answered the meeting doodle poll.